# Modelling impacts of spatially variable erosion drivers on suspended sediment dynamics

Giulia Battista[1], Peter Molnar[1], and Paolo Burlando[1]

[1]Institute of Environmental Engineering, ETH Zurich, 8093 Zurich, Switzerland

**Correspondence:** Giulia Battista (battista@ifu.baug.ethz.ch)

**Abstract.**

Suspended sediment load in rivers is highly uncertain because sediment production and transport at catchment scale are strongly variable in space and time, and affected by catchment hydrology, topography, and land cover. Among the main sources of this variability are the spatially distributed nature of overland flow as an erosion driver, and of surface erodibility given by soil type and vegetation cover distribution. Temporal variability mainly results from the time sequence of rainfall intensity during storms and snowmelt leading to soil saturation and overland flow.

We present a new spatially distributed soil erosion and suspended sediment transport module integrated into the computationally efficient physically based hydrological model TOPKAPI-ETH, with which we investigate the effects of the two erosion drivers - precipitation and surface erodibility - on catchment sediment fluxes in a typical pre-alpine mesoscale catchment. By conducting a series of numerical experiments, we quantify the impact of spatial variability of the two key erosion drivers on erosion-deposition patters, sediment delivery ratio, and catchment sediment yields.

Main findings are that the spatial variability of erosion drivers affects sediment yield by (i) increasing sediment production due to a spatially variable precipitation, while decreasing it due to a spatially variable surface erodibility, (ii) favoring the clustering of sediment source areas in space by surface runoff generation, and (iii) decreasing their connectivity to the river network by magnifying sediment buffers. The results highlight the importance of resolving spatial gradients controlling hydrology and sediment processes when modelling sediment dynamics at the mesoscale, in order to capture the key effects of sediment sources, buffers, and hillslope hydrological pathways in determining the sediment signal.

## 1 Introduction

Fine sediment produced in catchments by upland erosion and transported by rivers as suspended load is an important part of the global sediment budget (e.g. Peucker-Ehrenbrink, 2009) and an important driver of water quality and aquatic biota in rivers (e.g. Bilotta and Brazier, 2008). Human activity strongly interacts with the natural processes of suspended sediment production and transport, on the one hand by practices that enhance soil erosion, like agriculture, mining and deforestation, and on the other hand with the construction of sediment retention structures such as dams (e.g., Syvitski et al., 2005; Montgomery, 2007; Syvitski and Kettner, 2011; Borrelli et al., 2017). In the context of enhanced soil erosion, phenomena like the loss of soil productivity, the reduction of water quality due to higher turbidity and concentration of pollutants, and accelerated reservoir

siltation are expected (e.g. Pimentel et al., 1987; Davies-Colley and Smith, 2001). The combined effect of enhanced soil erosion and sediment retention by dams modifies the river sediment equilibrium and can result in river incision in the case of sediment starvation, contributing to undermine the stability of bridges and other infrastructures, and leading to coastal erosion (Kondolf, 1997; Chen and Zong, 1998; Schmidt and Wilcock, 2008). The opposite case of excessive sediment load in rivers may also lead to an increase in flood risk in alluvial floodplains due to sediment deposition (Yu, 2002; Walling, 2006; Rickenmann et al., 2016). The intensity of these effects is expected to grow in the future, as the magnitude and number of highly erosive extreme precipitation events are foreseen to increase in some parts of the world due to climate change and/or anthropic influence on land cover (e.g. Yang et al., 2003; Nearing et al., 2004; Peleg et al., 2019). Therefore, the monitoring and understanding of suspended sediment dynamics is essential to explain how disturbances produced by such human interventions may affect the sediment balance.

Fine sediment yield in rivers is usually estimated from intermittent measurements of sediment concentration by means of sediment-discharge rating curves (see Gao (2008) for a review). However, the development and use of these curves is often highly problematic because of the strong non-uniqueness of suspended sediment concentrations (SSCs), especially in small to medium sized catchment (up to 1000 km$^2$). Here, the same value of discharge (Q) often leads to a wide range of SSCs, producing highly scattered SSC-Q rating curves (e.g., Walling, 1977; Walling and Webb, 1982; Ferguson, 1986; Asselman, 2000; Horowitz, 2003). The strong variability in SSC is attributed to the high non-linearity of the sediment production and transport processes in time and space, and the presence of threshold and feedback mechanisms in sediment mobilization and transfer (e.g., Asselman, 1999; Collins and Walling, 2004; Seeger et al., 2004; Fryirs et al., 2007; Bracken et al., 2015).

Temporal and spatial variability in suspended sediment transport can originate from several sources (see Vercruysse et al. (2017) for a review). Among the sources of temporal variability, the role of hydrometeorological conditions (e.g. rainfall, antecedent wetness conditions, runoff) has been widely investigated, with a particular focus on the shape and direction of the hysteresis loops of the SSC-Q relation (Smith et al., 2003; Seeger et al., 2004; Zabaleta et al., 2007; Duvert et al., 2010; Dominic et al., 2015; Misset et al., 2019). Other sources of variability are the exhaustion of preferential sediment sources, the activation of new ones, and changes in the connectivity of such sources to the river network. These aspects have been studied for example as consequences of land use change and flow regulation (Olarieta et al., 1999; Siakeu et al., 2004; Costa et al., 2018). Variability of sediment transport in space depends on the distribution of sediment sources within the catchment, the catchment sediment connectivity, and the efficiency of sediment transport within the stream network. Wass and Leeks (1999) related differences in sediment loads across the basin to geomorphic and climatic gradients, while Fryirs and Brierley (1999) and Lang et al. (2003) reconstructed the change of sediment sources in time and their coupling with the channels. The problem of catchment sediment connectivity has been addressed from a conceptual point of view, by introducing the ideas of structural and functional connectivity, to distinguish between the physical connection among landscape units and the connectivity generated by the system process interactions (Wainwright et al., 2011; Fryirs, 2013; Bracken et al., 2015). Based on these concepts, several indices have been introduced to assess sediment connectivity in a river basin (see Heckmann et al. (2018) for a review).

The above studies highlight the need to account for both types of variability (temporal and spatial) in order to investigate basin sediment dynamics. Including this variability is especially important at the medium and large catchment scale and in

mountainous environments, where the gradients of climatic and physiographic variables are most relevant. Few studies have focused specifically on the impacts of spatially variable erosion drivers on suspended sediment dynamics in such environments. A systematic investigation of this research gap can be performed by means of numerical models that include the main hydrological processes, their temporal dynamics and distribution in space, as well as their interaction with the topography and morphology of the basin. Several existing models are partially suitable for this task. The main limitations are that many are only suitable for event-applications (Answers (Beasley et al., 1980), KINEROS (Woolhiser et al., 1990), WEPP (Nearing et al., 1989)) or present simplified hillslope hydrology and runoff formation solutions, as in the case of WATEM/SEDEM (Van Rompaey et al., 2001), landscape evolution models, e.g. Caesar-Lisflood (Coulthard et al., 2013), SIBERIA (Hancock et al., 2000), or some large-scale sediment flux models, e.g. WBMsed (Cohen et al., 2013) and Pelletier (2012). More suitable approaches are tRIBS (Francipane et al., 2012), which includes a physically based hydrological component suitable for long-term process simulations, and DSHVM (Doten et al., 2006), which features a detailed hydrology-vegetation component and sediment module. However, the number of processes represented in these two models requires a high computational power and their applications have so far been limited to small basins and/or short time scales. Finally, Tsuruta et al. (2018) present a spatially distributed model especially for large basins, which, being based on a land-surface model, features an approximated coarse-scale representation of hydrological and sediment connectivity on the hillslopes.

In this work, we present a modelling approach especially suitable for alpine catchments with highly variable climate and complex topography, that integrates a new spatially distributed soil erosion and suspended sediment transport module within the computationally efficient, physically based hydrological model TOPKAPI-ETH (Fatichi et al., 2015). The model combines unsteady simulation of surface and subsurface water fluxes with a simple hillslope erosion and sediment transport component. The sediment component is simple by design, to avoid over-parameterization and to maintain computational efficiency enabling applications to medium and large catchments. The model allows continuous high spatial resolution ($\Delta x$=100 m) simulations to track overland flow and hillslope sediment transport by local changes in soil moisture dynamics produced by rainfall, snowmelt and lateral drainage over long periods of time. The model also allows high temporal resolution ($\Delta t$=1 hr) simulations to capture fast runoff response to the hydrological drivers, which, together with the topographically driven flow routing, reproduces the connectivity of water and sediment pathways in the catchment over time. The combined hydrology-sediment model is unique in its process completeness and applicability to mesoscale catchment simulation at high resolutions, compared to most other approaches.

The overall aim of this research is to provide a state-of-the-art catchment hydrology-sediment modelling framework to better understand the sources of variability in suspended sediment concentrations and their effects on predictions of sediment yield. Accordingly, we conducted numerical experiments on a mesoscale pre-alpine river basin, where we turned on and off the spatial variability in two key erosion drivers - rainfall and surface erodibility - to quantify their individual and combined effect on suspended sediment mobilization and transfer. We address the following specific research questions: (RQ1) Does fully distributed physically-based hydrology-sediment modelling predict variability in SSC-Q relations that is in agreement with observations? We argue which key hydrological processes are needed in such a model and why. (RQ2) Can we identify the location of sediment sources and quantify their productivity and connectivity with such a modelling approach? We assess

the effect of the spatial distribution of rainfall and surface erodibility on hillslope erosion-deposition patterns and sediment mobilization, and we quantify the sediment source connectivity to the river network by analysing the sediment delivery ratio along the main stream and in tributary basins. (RQ3) Is the effect of spatially distributed erosion drivers visible in sediment yield at the catchment outlet? We show how integration of the spatially variable inputs in space impacts sediment yield under different scenarios.

## 2 Methods

### 2.1 Hydrology-sediment model description

The model we present in this work is an extension of the hydrological model TOPKAPI-ETH (Fatichi et al., 2015), which we integrated with a new hillslope erosion and channel suspended sediment flow module. The TOPKAPI-ETH hydrological model was chosen because of its spatially distributed nature and physically based representation of the major hydrological processes, combined with a reasonable computational demand. The model is based on a regular square grid discretization in space and a 3-layer vertical discretization of the subsurface. The river network is identified in the domain by means of a flow accumulation algorithm based on the topography. The transition between hillslope and channel process description, i.e. the beginning of the model river network, is set by a user-defined critical upstream area, or river initiation threshold RT, above which water flow is modelled as channel flow. Each river network cell can be fully or partially covered by the stream, depending on the actual stream width and grid cell resolution.

In TOPKAPI-ETH surface and subsurface flow is simulated by the kinematic wave approximation, with resistance to flow given by surface roughness and soil transmissivity as a function of soil properties. Water may saturate the soil locally and lead to overland flow generation by saturation excess or by infiltration excess in case of high rainfall intensities. Soil is dried by evapotranspiration, lateral drainage and percolation to groundwater storage. The model includes snow cover accumulation and melt, which are important in the water balance of alpine basins. For further details about the model see Fatichi et al. (2015). TOPKAPI-ETH allows long-term, high resolution simulations (time step $\Delta t$=1hr, grid size $\Delta x$=100 m) in medium and large catchments (>1000 km$^2$), even when integrated with a sediment mobilization and transfer component, since the kinematic wave approximation of the surface and subsurface flow routing are solved analytically (Liu and Todini, 2002).

In the new sediment module of TOPKAPI-ETH, the mobilization and routing of fine sediment on the hillslopes takes place by action of overland flow, which is assumed to transport sediment at its maximum capacity. As a consequence, deposition and erosion can occur on the hillslopes at a rate $D$ [kg m$^{-3}$ s$^{-1}$] depending on the hydraulic and topographic properties of the cells along the flow path:

$$D = \nabla \cdot q_s, \tag{1}$$

where $q_s$ [kg m$^{-2}$ s$^{-1}$] is the overland flow transport capacity, modelled following Prosser and Rustomji (2000) as a function of the specific overland flow discharge $q$ [m$^2$ s$^{-1}$] and the surface slope $S$ [m/m]:

$$q_s = \alpha q^\beta S^\gamma, \tag{2}$$

where $\beta$ and $\gamma$ are transport exponents, and $\alpha$ [kg s$^{0.4}$ m$^{-4.8}$] is a calibration parameter that captures the effect of land surface and soil properties on erosion and sediment transport. The sediment flux $q_s$ is directed to the downstream cell with the steepest gradient. Sediment inflow into a cell can be from one or more upstream cells. Once the sediment mobilized and routed on the hillslopes reaches the channel, it is assumed to move as suspended sediment load.

The suspended sediment flux in the river network is treated as an advection process and solved with the same numerical methods used for water flow. The 1D equation of suspended sediment flux in the channel, integrated over the river cross-section, is:

$$\frac{\partial AC}{\partial t} = E - \frac{\partial QC}{\partial x}, \tag{3}$$

where $Q$ [m$^3$ s$^{-1}$] is the river discharge, $C$ [g m$^{-3}$] is the SSC, $A$ [m$^2$] is the cross-section area of flow and $E$ [g m$^{-1}$ s$^{-1}$] represents the exchange of sediment with the bed and local sediment sources. By following the reasoning of Liu and Todini (2002), Equation 3 can be integrated along the length of the grid cell (i.e. in the flow direction), within which the values of the variables are assumed to be constant, and then solved analytically as a first-order ordinary differential equation:

$$\frac{\partial V_i C_i}{\partial t} = E_i X + Q_{in} C_{in} - \frac{U_i}{X} C_i V_i, \tag{4}$$

where $X$ [m] is the length of the grid cell size, $V_i$ [m$^3$] the volume of water inside a cell ($V_i = A_i X_i$), $U_i$ [m s$^{-1}$] the mean flow velocity, $C_i$ and $E_i$ are the mean values of $C$ and $E$ inside the grid-cell. $Q_{in}$ and $C_{in}$ are the discharge and sediment concentration entering the cell $i$ from the upstream grid cell $(i-1)$.

## 2.2 Study site

We chose to investigate the research questions outlined above on the Kleine Emme river basin, a pre-alpine catchment located in central Switzerland. Here the natural regime of water and sediment flow is almost unaltered, and the basin is sufficiently large for spatial variability in erosion drivers to have an impact. The basin has an area of 477 km$^2$, an elevation range of 430-2300 m. a.s.l. and a mean annual precipitation of 1650 mm (Figure 1a). The mean annual discharge at the outlet is 12.6 m$^3$/s. The catchment is mostly natural, with more than 50% of the surface covered by forest and grassland (Figure 1c). No use of water for irrigation or hydropower is known and significant sediment-retaining infrastructures are absent. Moreover, the absence of glaciers means that fine sediment production in the basin is mostly driven by overland flow and rainfall processes. Finally, the diverse geomorphology of the basin has been the subject of several studies and long-term estimates of denudation rates are available (e.g., Schlunegger and Schneider, 2005; Schwab et al., 2008; Dürst Stucki et al., 2012; Van Den Berg et al., 2012; Clapuyt et al., 2019).

Measurements of precipitation, air temperature and sunshine duration are available from automatic weather stations located inside or in the vicinity of the basin operated by MeteoSwiss. The information about the spatial distribution of precipitation inside the basin is available from the 1x1 km daily gridded product of MeteoSwiss RhiresD (Frei and Schär, 1998; Schwarb, 2000). Streamflow is monitored at Werthenstein and at the basin outlet by the Federal Office of the Environment (FOEN) and at Sörenberg by the Canton Luzern (Figure 1a). FOEN also provided the cross section measurements for the main channel

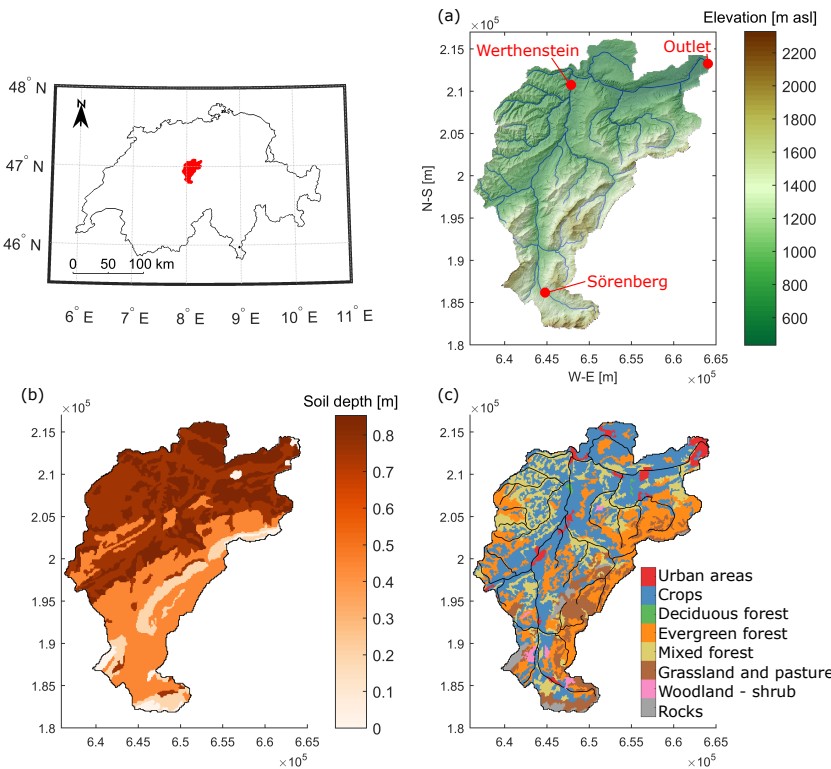

**Figure 1.** (a) Digital Elevation Model (DEM) of the Kleine Emme basin and location of discharge gauges (source SwissAlti3D, 2017), (b) Soil depth, derived from the Swiss soil map (Bodeneignungskarte, 2012) and (c) Land cover derived from Corine Land Cover map (CLC, 2014). The coordinate system is CH1903.

of the river and measurements of suspended sediment concentration. SSCs have been manually sampled at the outlet since 1974, but with a regular frequency of two samples a week only since 2004. Because of the low temporal resolution of these measurements, which is typical of many river sediment monitoring networks, we expect this dataset to miss extreme SSCs generated by flood events or very localized sediment sources. Finally, the information about soil type and depth for the basin is available from the soil map of Switzerland (Bodeneignungskarte, 2012) (Figure 1b) and land cover is provided by the Corine Land Cover dataset (Figure 1c).

### 2.3 Model setup and calibration

#### 2.3.1 Hydrology

Given the period of availability of suspended sediment measurements in the Kleine Emme, the simulation was set up for the years 2003 to 2016, where the first year is considered a warm-up period. The meteorological input data required by the hydrological component of TOPKAPI-ETH are hourly precipitation, air temperature and cloud cover. The precipitation input

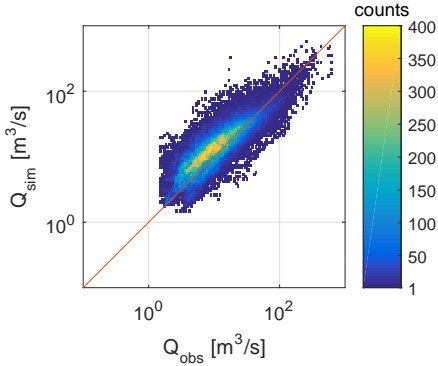

**Figure 2.** Performance of the hydrological model: density plot of observed vs simulated hourly discharges at the outlet of the river basin for the period 2004-2016.

file was created by combining station and gridded precipitation datasets following the approach of Paschalis et al. (2014). In this approach hourly precipitation measured at the rain gauges was spatially interpolated to match the spatial distribution of the daily precipitation in the gridded RhiresD dataset. The hourly time series of measured air temperature were extrapolated across the model domain to different elevations with a temperature lapse rate of -5.5 °C/km. Cloud cover transmissivity was derived from the hourly sunshine duration measurements following the empirical relation proposed by Kasten and Czeplak (1980).

The model was run at a $\Delta x$=100 m spatial resolution and a constant time step $\Delta t$=1 hour. To initiate the model calibration, realistic values of the hydrological parameters were assigned based on the soil characteristics and previous investigations (Paschalis et al., 2014; Pappas et al., 2015). The soil hydraulic conductivity and the residual and saturation soil water content parameters were then adjusted in order to maximize the performance of the hydrological model in terms of correlation coefficient (r), Nash-Sutcliffe efficiency (NSE) and root mean square error (RMSE) for discharge measured at three streamflow gauging stations.

The final configuration of the hydrological model performed very well in reproducing the observed discharge at the outlet and at Werthenstein (see Table 1 and Figure 2). Discharge data are available at a sub-daily resolution at Sörenberg only from the year 2005; therefore, the evaluation of the performance at this station does not consider the first year of simulation. The model performance at this station is slightly worse, probably also due to the lower accuracy of the measurements, but still satisfactory.

### 2.3.2 Setup of the sediment module

The inputs needed to run the hillslope erosion and suspended sediment transport modules are the parameters $\alpha$, $\beta$ and $\gamma$ in Equation 2. The $\beta$ and $\gamma$ parameters are assumed spatially uniform and equal to $1.4$, following Prosser and Rustomji (2000). The parameter $\alpha$ contains information about the soil and land surface properties that influence the rate of soil erosion. We derived the spatial distribution of $\alpha$ by the product of the soil erodibility parameter $K$ of the Universal Soil Loss equation (USLE), computed for Switzerland by Schmidt et al. (2018), and the land use USLE parameter $C$, which we derived from

**Table 1.** Hydrological performance for the simulation period 2004-2016 at the three flow monitoring stations in terms of correlation coefficient (r), Nash-Sutcliffe efficiency (NSE) and root mean square error (RMSE) for data simulated at the hourly resolution and aggregated to daily, monthly and annual values.

| | Outlet | | | Werthenstein | | | Sörenberg (2005-16) | | |
|---|---|---|---|---|---|---|---|---|---|
| | r | NSE | RMSE | r | NSE | RMSE | r | NSE | RMSE |
| | [-] | [-] | $[m^3/s]$ | [-] | [-] | $[m^3/s]$ | [-] | [-] | $[m^3/s]$ |
| Hour | 0.84 | 0.69 | 0.75 | 0.84 | 0.65 | 0.74 | 0.63 | 0.72 | 1.43 |
| Day | 0.91 | 0.80 | 0.53 | 0.90 | 0.78 | 0.52 | 0.80 | 0.56 | 0.83 |
| Month | 0.93 | 0.76 | 0.28 | 0.92 | 0.77 | 0.26 | 0.88 | 0.77 | 0.38 |
| Year | 0.93 | - | 0.18 | 0.92 | - | 0.13 | 0.79 | - | 0.10 |

Yang et al. (2003) (see Figure S1). In this way we implicitly account for the influence of particle size distribution, organic matter content, soil structure, permeability, surface roughness and vegetation cover in determining the spatial distribution of surface erodibility. A comparable approach is proposed by Hancock et al. (2017).

The ratio between the product of $C$ and $K$ of the different classes was then kept constant in the calibration process and $\alpha$ was calibrated by multiplying the $CK$ values by a spatially constant parameter $\alpha_1$:

$$\alpha(x,y) = \alpha_1 C(x,y) K(x,y), \tag{5}$$

where $x$ and $y$ are coordinates in space. With respect to channel processes, the water column-bed exchange and local sediment source term $E$ in equation 3 is unknown. In the Kleine Emme significant deposits of fine sediment in the river bed are not present and bedrock is often exposed, indicating an efficient fine sediment transport downstream (Schwab et al., 2008). Furthermore, the infrequent SSC measurements do not allow to quantify the term explicitly. This leads us to assume that $E$=0 for this river. However, by setting $E$=0 we neglect also local sediment sources along the channels, which is probably an approximation of the sediment production processes in this case study. Also on the hillslopes, localized sediment sources are not explicitly modelled and are present only insofar they are represented by high $C$ and $K$ values. The lack of explicit inclusion of point sediment sources and their modelling is a limitation of the current approach, which we will address in future work.

### 2.3.3 Calibration of the sediment module

We found that the parameters that have the highest influence on matching the observed and simulated SSC at the outlet are the river initiation threshold $RT$, i.e. the extension of the modelled river network, and the $\alpha_1$ constant, defining the soil erodibility. RT has a small influence on discharge, as shown by Table S2, while it is a relevant parameter for the modelling of hillslope erosion and sediment transport. Since fine sediment mobilization can only take place on the hillslopes, the extension

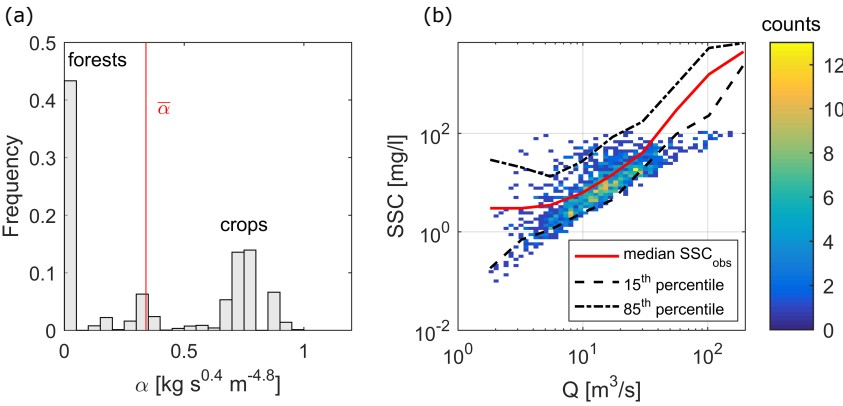

**Figure 3.** (a) Frequency distribution of the calibrated surface erodibility parameter $\alpha$, with mean $\alpha$ indicated with the red line; (b) density plot of the simulated SSC at outlet compared with measurements, the lines show the median (red) and $15^{th}$ and $85^{th}$ percentile (black dashed) of the observations.

of the channels onto the hillslopes influences the magnitude of the sediment input into first-order channels and subsequently downstream through the river network.

In the calibration of the model we focused on measurements below the $85^{th}$ percentile, because flood events in the SSC data
are likely under-sampled, due to the monitoring strategy, and the model is expected to underestimate the SSC extremes due to the simplified representation of the sediment mobilization processes. The calibration was performed by matching the trend and the dispersion of the measured and modelled SSC-Q cloud of points. This was done by visual matching and by comparing the mean and variance of the observed SSCs. The final calibrated parameters are $\alpha_1$=0.0138 kg m$^{-1.8}$ s$^{-2.6}$ and $RT$=0.4 km$^2$. The histogram of $\alpha$ and its spatial distribution are shown in Figures 3a and 7d, respectively; the spatial mean of $\alpha$ is $0.3412$
220 kg s$^{0.4}$ m$^{-4.8}$. We note that the calibrated river initiation threshold is very close to the drainage area that Schlunegger and Schneider (2005) propose as the threshold area at which channelized processes start dominating over hillslope processes in the development of the landscape in this study basin (0.1-0.2 km$^2$).

Using this parameterization, the measured SSC-Q cloud of points is captured very well for moderate discharges (Figure 3b), whereas the concentrations at highest discharges are underestimated, as expected. Overall, 90.4% of the simulated SSCs fall
within the $5^{th}$ and $95^{th}$ percentile of the observations and, if the simulated SSCs are sampled at the hours of observations and compared to the observations limited to their $85^{th}$ percentile, the observed SSC mean and variance are reproduced with very small errors ($\overline{SSC}_{sim} = 12.40 \; mg/l$, $\overline{SSC}_{obs} = 12.20 \; mg/l$; $\sigma^2_{sim} = 210.47 \; mg/l$, $\sigma^2_{obs} = 233.15 \; mg/l$) (Figure S2). We attribute the underestimation of high sediment concentrations (above $85^{th}$ percentile) to missing localized sediment sources, i.e. mass wasting processes in the model, which are responsible for point sediment sources, like landslides, debris flows and
bank erosion. Further evaluation of the suspended sediment module performance can be found in Table S1 and Figure S3.

## 3   Erosion driver numerical experiments

In order to investigate the processes leading to the scatter in the SSC-Q relation and how they affect the spatial organization of sediment transport, we performed simulation experiments that quantify the role of spatial variability in two key erosion drivers - precipitation and surface erodibility. Precipitation is the main hydrological driver of hillslope erosion through the overland flow term $q^{\beta}$ in Equation 2, while surface erodibility is represented by the parameter $\alpha$ in Equation 2.

We designed four numerical experiments by combining spatially variable and/or uniform distributions of the two erosion drivers (Figure 4). The reference experiment (SIM 1) accounts for the highest level of complexity by considering both precipitation and erodibility variable in space. This is the experiment with which the model was calibrated (see section 2.3.3). The second experiment (SIM 2) aims to quantify the role of the spatial variability in precipitation, by reducing it to be uniformly distributed in space. The temporal variability was preserved by setting the hourly precipitation in each cell equal to the mean hourly precipitation over the catchment. The third experiment (SIM 3) is designed to investigate the role of the spatial variability in surface erodibility by reducing it to uniform surface erodibility throughout the basin, equal to the mean value of the calibrated spatial distribution of $\alpha$. A fourth experiment (SIM 4), where the spatial variability in both drivers was reduced to uniform, was run to quantify the combined effect of the two erosion drivers.

## 4   Results

In section 4.1 we evaluate the spatio-temporal variability in sediment mobilization and transport and the scatter of the SSC-Q relation it produces by the fully distributed erosion drivers in SIM 1 (RQ 1). The spatial distribution of suspended sediment transport is then evaluated in subsequent sections and related to the hydrological response of the basin (RQ 2). We compare the activation of sediment sources and the sediment mobilization in the four simulations (section 4.2) and we quantify the connectivity of sediment transfer by means of the sediment delivery ratio (section 4.3). Finally, in section 4.4, we analyze the sediment load at the outlet as a function of the sediment spatial properties observed in the different scenarios (RQ3).

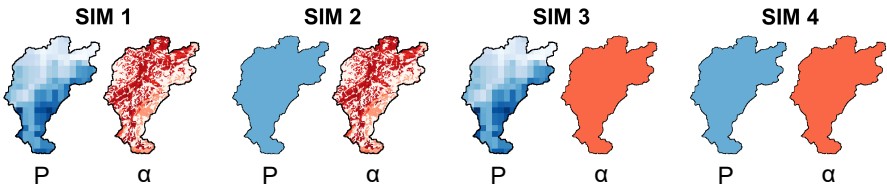

**Figure 4.** Summary of model runs: in SIM 1 sediment mobilization and transfer are driven by a spatially distributed precipitation (P) and surface erodibility ($\alpha$), in SIM 2 and SIM 3 the spatial variability in precipitation and surface erodibility have been removed, respectively, and in SIM 4 both spatial variabilities have been removed.

## 4.1 Spatio-temporal variability in erosion and sediment transport

The modelled scatter in the SSC-Q relation in SIM 1 explains about 30% of the measured concentration range for discharges up to the $85^{th}$ percentile, while it shows a much more significant underestimation for the highest flows (Figure 3b). For a comparison between the SSC-Q scatter generated by the different scenarios of erosion drivers, the reader is referred to Figures S5 and S6. In the following we analyse the sources of this variability, by showing the time series of discharge and the sediment load and concentration for one representative year (Figure 5a) and by analysing the pattern of erosion and deposition across the basin from the entire simulation period (Figure 5b).

High sediment fluxes in April and May, which are evident both in observations and in the model (Figure 5), indicate the contribution of snowmelt to discharge and the erosion of the surface by widespread overland flow. Summer events (storms) provide a small contribution to the yearly sediment yield. However, they generate some of the highest sediment concentrations in the model even though the runoff remains low. As expected, high SSCs are not observed in the measurements during summer, because sediment is rarely sampled during summer floods (see section 2.2). In winter months, snow covers the majority of the catchment and maintains the sediment flux very close to zero in both observations and simulations (Figure 5a).

Most of the erosion is simulated in the south-eastern part of the basin, where slopes are steeper, soil is thinner and the highest precipitation, snow accumulation and melt occur (Figure 5b). In these regions, it is easier to saturate the soil and generate runoff over larger areas that merge and generate connected areas of overland flow, thus producing wide erosional surfaces on steep mountain flanks. Deposition is simulated at the valley bottoms or at locations of slope reduction. In the north-western part of

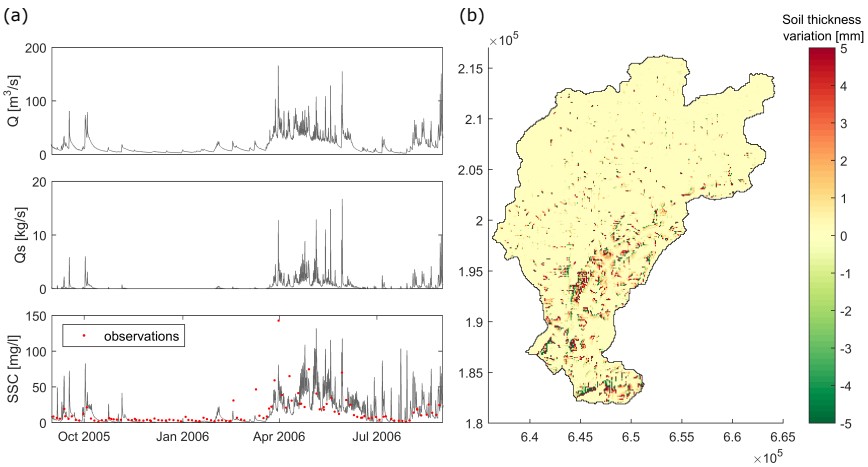

**Figure 5.** (a) Time series of hourly modelled discharge Q, suspended sediment load Qs and concentrations SSC for one year at the outlet. The red dots in the SSC time plot show the observed values. (b) Change in soil thickness at the end of the 13-year simulation. Positive values indicate erosion, negative values indicate deposition.

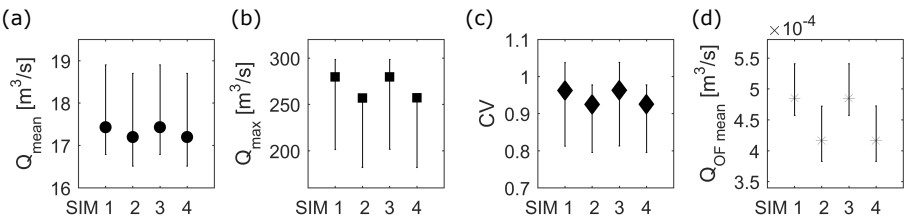

**Figure 6.** Comparison of the hydrological response of the basin in the four simulations: (a) mean annual discharge $Q_{mean}$, (b) annual flood $Q_{max}$ and (c) the coefficient of variation CV of the hourly discharge at the basin outlet, and (d) the mean annual overland flow runoff over the basin $Q_{OFmean}$. The markers indicate the mean values and the lines the interval between the $25^{th}$ and $75^{th}$ percentile of the distribution from hourly data over the entire simulation period.

the basin, overland flow remains constrained to the channel headwaters due to the deeper soil and to the higher drainage density
of the area. This distribution of erosion is coherent with the different geomorphological characteristics of the two areas of the basin, as further discussed in section 5.2. We observe that, because of the transport capacity approach in the hillslope transport module, areas of strong erosion are often associated with significant deposition downstream. In the following, we will refer to these areas of strong erosion as sediment source areas.

The mean annual suspended sediment load generated by SIM 1 is $1.42 \ 10^4$ t/y, which is significantly lower than the $2.83 \ 10^5$
t/y computed from the measurements at Littau by Hinderer et al. (2013). Consistently, the mean annual erosion rate of $0.07$ mm/y underestimates the denudation rates derived from $^{10}$Be samples in the Entlen and Fontanne sub-basins by Wittmann et al. (2007), Norton et al. (2008) and Van Den Berg et al. (2012) (between 0.38 and 0.52 mm/y), which are from active erosion areas and integrate over a much longer time span of about $10^4$ years. The lower estimates of sediment load and erosion rates by our model compared to such data is expected, given the underestimation of SSC at high flows by the model. This limitation
will be further discussed in section 5.1.

### 4.2 Sediment sources and sediment production

To interpret the effect of the spatial variability of precipitation and surface erodibility on sediment transport, in Figure 6 we compare the hydrological response of the basin in the four simulations in terms of the mean annual discharge $Q_{mean}$, annual flood $Q_{max}$, coefficient of variation CV of the hourly discharge at the basin outlet, and mean annual overland flow runoff over
the basin $Q_{OFmean}$. Figure 6 indicates that uniform precipitation (SIM 2 and 4) is less efficient in producing runoff ($Q_{mean}$, $Q_{max}$ and $Q_{OFmean}$) and therefore has a lower erosive power. Spatially variable precipitation (SIM 1 and 3) produces a greater flow variability, because it allows to distinguish between convective rainfall patterns, which affect smaller regions of the basin, and stratiform rainfall patterns which affect the entire basin with lower precipitation intensities.

The sediment response of the basin in the four simulations is compared in the following by looking at the distribution of
sediment source areas and their productivity. Figure 7 compares soil thickness variation in SIM 2 and 3 respectively to SIM 1. Figures 7b and 7c show the difference between the variable and uniform precipitation maps for erosion and deposition,

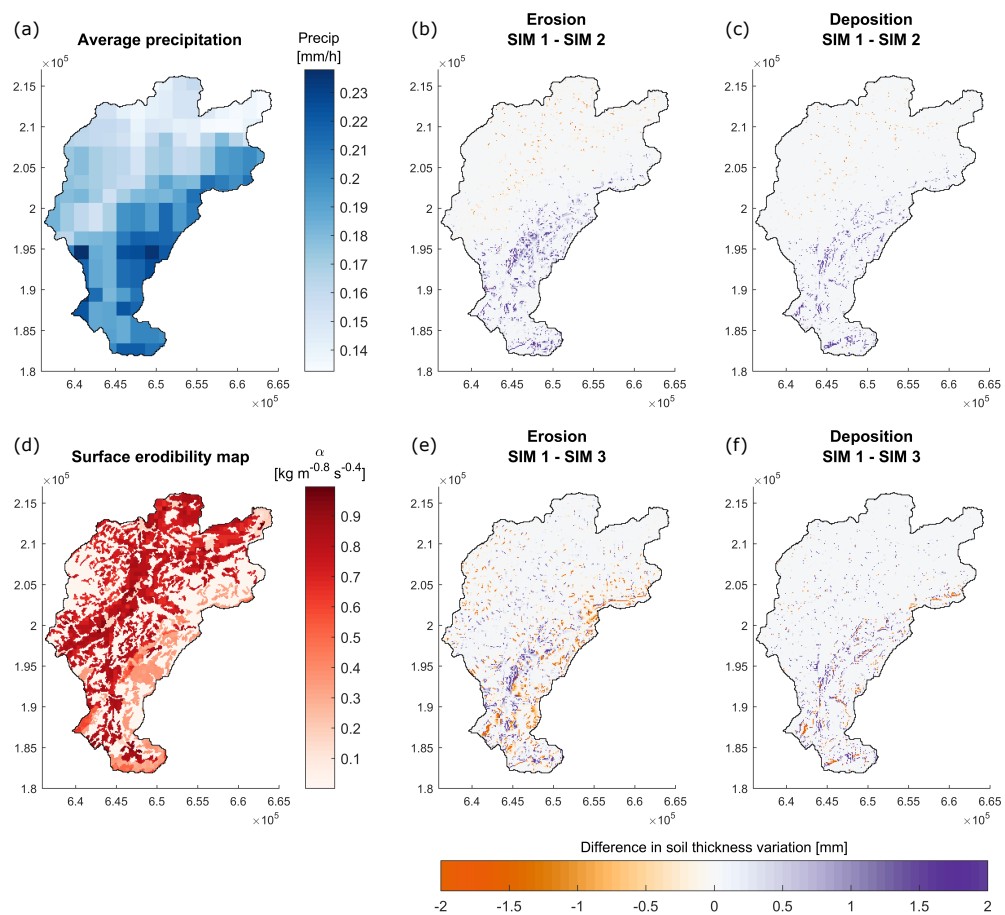

**Figure 7.** (a) Average spatial distribution of precipitation intensity for the period 2004-2016, (b, c) difference between erosion/deposition generated by variable and uniform rainfall in 13 years, (d) spatial distribution of calibrated surface erodibility $\alpha$, (e, f) difference between erosion/deposition generated by variable and uniform surface erodibility in 13 years. A positive value indicates more erosion/less deposition by variable precipitation or surface erodibility and a negative value indicates less erosion/more deposition.

respectively. Similarly, Figures 7e and 7f show the difference between the variable and uniform surface erodibility maps for erosion and deposition separately. A positive value indicates more erosion/less deposition by variable precipitation or surface erodibility, and a negative value indicates less erosion/more deposition.

The results show that with uniform precipitation, erosion and deposition are reduced in the south-eastern part of the basin and increased in the north-western (Figure 7b and 7c). The overall patterns reflect the average spatial distribution of precipitation in the Kleine Emme catchment for the years 2004-2016, with the highest mean rain intensities associated with more erosion (Figure 7a). Uniform surface erodibility increases sediment erosion and deposition in the forested areas and reduces them in crop areas (Figure 7e and 7f). In both cases, the overall effect of removing the spatial variability in erosion drivers is a more

uniform distribution of the sediment source areas across the basin.

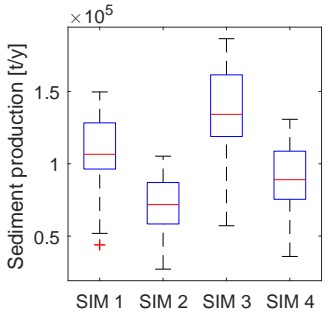

**Figure 8.** Sediment production in the basin as total sediment detached annually for the four simulations. Boxplots (median, interquartile range and outliers) show the interannual variability in the period 2004-2016.

To quantify the erosional power of the four combinations of erosion drivers, we computed the total sediment mass detached yearly across the whole basin (referred to as sediment production) in the four simulations. The distribution of the yearly sediment production with interannual variability is reported in Figure 8. We observe that the removal of spatial variability generates two opposite effects for precipitation and surface erodibility. Sediment production increases when removing the

spatial variability in surface erodibility and decreases when removing the spatial variability in precipitation, coherently with the reduced erosive power observed in Figure 6. In SIM 4 the balance between the two opposing effects determines a slight overall reduction in sediment production. The differences between the scenarios are within natural interannual variability in sediment production, but they are all statistically significant for change in median.

### 4.3   Connectivity of sediment transfer

The connectivity of sediment transfer, i.e sediment source areas linked to the river network, within the catchment for the different simulation configurations has been quantified by means of the sediment delivery ratio (SDR). The SDR is defined according to Walling (1983) as the ratio of the sediment delivered at the outlet of a selected area to the gross erosion in that area. The mean annual SDRs, which were computed at the outlet point of the main tributaries and at several cross-sections along the main channel, are reported in Figure 9 as a function of the drainage area.

Sediment connectivity along the main channel shows an increasing trend as a function of the upstream area for all simulations (Figure 9c). This trend is explained by the higher SDR of the tributaries compared to that of the main channel (Figure 9b) and by the absence of significant sediment sinks in the main channel. For the subbasins with outlets along the main channel, removing the spatial variability in surface erodibility (SIM 3) has the overall effect to increase sediment connectivity. In some tributaries, however, the opposite effect is observed (T5 and T6). Finally, Figure 9c shows that removing the spatial variability

in precipitation (SIM 2 and 4) also increases the SDR, therefore sediment connectivity (compared to SIM 1 and 3, respectively).

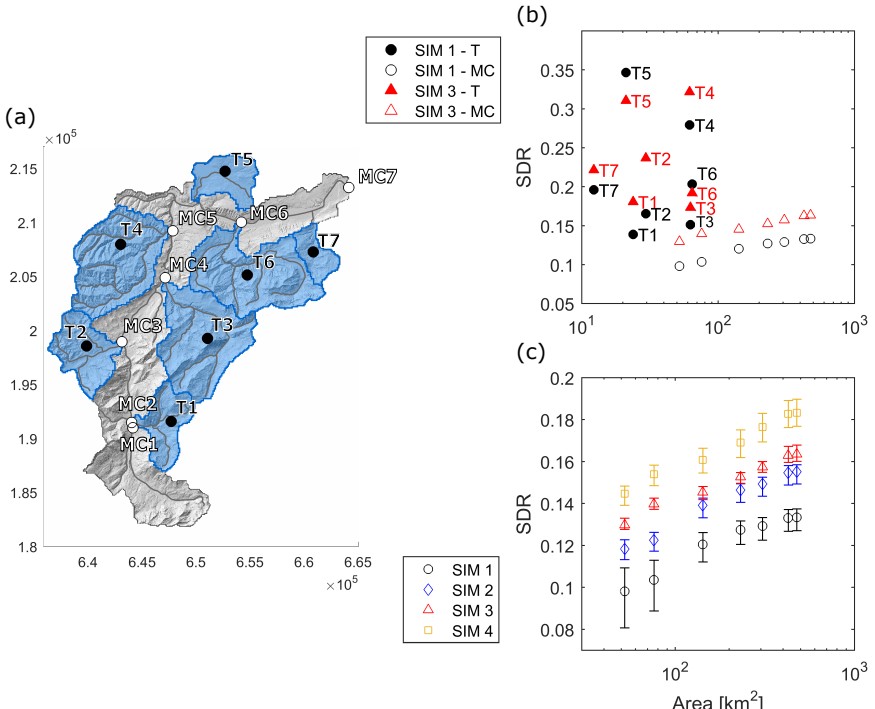

**Figure 9.** (a) Locations where the sediment delivery ratio has been computed: at the outlet of the main tributaries (T) and along the main channel (MC), (b) mean annual SDR vs drainage area for tributaries and points along the main channel for distributed rainfall simulations, (c) comparison of mean annual SDRs at the main channel points for the four simulations. The error bars show the interquartile range of the annual SDR variability.

## 4.4 Sediment loads and initial soil moisture

The distribution of annual sediment yields at the outlet generated by the four simulation experiments showed that distributed precipitation simulations (SIM 1 and 3) generated higher sediment loads than their uniform precipitation equivalents (SIM 2 and 4) (Figure 10a). Distributed erodibility (SIM 1 and 2) produced smaller sediment loads than uniform erodibility (SIM 3 and 4).

To further investigate the differences among the sediment yield distributions, in Figure 10b we show the influence of spatial variability in rainfall and surface erodibility on event-based sediment yields for high and low initial soil moisture ($SM_0$) conditions. After separating the outlet hydrograph into single events, we computed the total sediment yields for each event and compared the distributions of the events with high and low initial soil moisture. Low $SM_0$ events are defined as those with catchment-averaged $SM_0$ smaller than the $20^{th}$ percentile of the $SM_0$ distribution; high $SM_0$ events have a $SM_0$ greater than the $80^{th}$ percentile. The hydrological model performance for these events is good and comparable to the entire simulation performance, however it indicates a tendency to overestimate especially for low $SM_0$ events (see Table S3 and Figure S4).

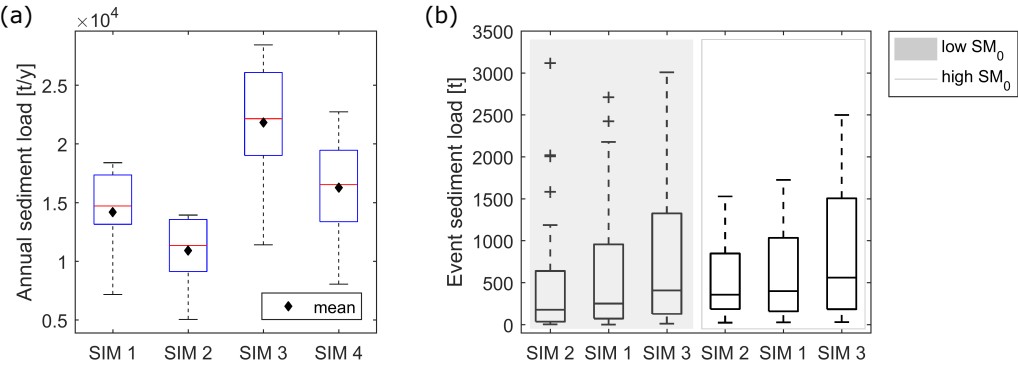

**Figure 10.** (a) Boxplots of annual sediment load and their mean values at the outlet of the catchment in the four simulation experiments, (b) boxplots of event sediment loads divided into low and high initial soil moisture conditions. The boxplots compare the effect of the spatial variability in precipitation and surface erodibility on events with different initial soil moisture.

The distributions of event sediment yields largely overlap, however it is possible to observe that sediment yield is more affected by the precipitation spatial variability when $SM_0$ is low. The differences between the median, $25^{th}$ and $75^{th}$ percentile of the SIM 1 and 2 are bigger for low $SM_0$ than for high $SM_0$. On the contrary, removing variability in surface erodibility seems to equally affect low and high initial $SM_0$ events (Figure 10b).

## 5 Discussion

### 5.1 Sources of concentration variability

The modelling approach presented here can reproduce part of the observed SSC-Q scatter, implying that it contains some of the relevant sources of sediment concentration variability in the hydrological and sediment production processes at the catchment scale (Figure 3b). However, it also highlights that to fully capture the scatter, other sources should be included. The comparison of simulated and observed hourly SSC is satisfactory (Figure S3).

The sources of variability accounted for by the deterministic modelling of the hydrology and sediment transfer are the time-varying meteorological inputs and the spatially distributed nature of the model. The precipitation input combines both temporal and spatial components of variability. The temporal component is visible in Figure 5a, showing that the same sediment concentration can correspond to a large range of discharge values, depending on the type of event and the initial soil wetness conditions that precede it. Spatial variability in precipitation contributes to the SSC-Q scatter, by increasing the flow variability itself (Figure 6c) and by allowing the same discharge at the outlet to be generated by many combinations of overland flow situations over the hillslopes. Each of these combinations activates different sediment sources that have a characteristic hydrological and sediment signal and connectivity to the river network. In particular, we identify localized high-intensity summer storms as a main source of scatter, while snowmelt and winter storms produce a more homogeneous response throughout the

basin. The spatially variable surface erodibility can additionally contribute to the uniqueness of the sediment signals of the activated source areas, when its spatial distribution is such to enhance the topographic heterogeneity within the basin.

Other sources of variability in sediment transport are implicit in the spatially distributed nature of the model, which allows to account for the heterogeneity of topography, soil depth and soil properties at very high resolution. These heterogeneities are responsible for the residual scatter of SIM 4, where the variability of both erosion drivers have been removed.

It is worth noting that, because the sediment storage on hillslope cells is not exhausted during our simulation experiments, sediment availability does not influence sediment production in our study. Therefore, sediment availability in the simulation experiments does not drive changes in the dominant sediment sources and does not add spatial variability to the sediment response.

The main limitation of our approach in reproducing SSC variability is, however, the lack of processes representing very localized sediment sources, which are usually characterized by a threshold behavior and therefore diversify the local sediment response. In this respect, Schwab et al. (2008) showed that in the Kleine Emme basin short timescale threshold processes are responsible for the export of regolith produced by soil creep in landslides. The absence of these processes in our model is likely one of the main reasons for the smaller-than-observed modelled SSC-Q scatter, but also for the underestimation of the highest SSCs, the soil erosion rate and annual sediment load, presented in section 4.1. Finally, we acknowledge that also inherent stochasticity in the sediment mobilization and transfer are responsible for part of the observed SSC-Q rating curve scatter (e.g. Fuller et al., 2003; Malmon et al., 2003). This inherent stochasticity cannot be reproduced by our modelling approach with deterministic simulation, but it can be included with stochastic simulation experiments and a probabilistic framework (e.g. Bennett et al., 2014). We are working on overcoming these limitations in future research.

### 5.2 Spatial organization of suspended sediment transport

The explicit combination of hydrological processes and topographic and land use effects in the model can help to investigate the spatial organization of sediment transport, and in particular, how this is affected by the spatial variability in erosion drivers. Spatial variability enhances the heterogeneity of erosion and deposition across the catchment, thus favoring the clustering of sediment source areas (Figure 7). Sediment production is increased by the spatially variable precipitation (SIM 1 and SIM 3), due to increased erosive power (Figure 8). The effect of a spatially variable surface erodibility depends on the distribution of overland flow relative to that of surface erodibility and, in this case, the lower sediment productions of SIM 1 and 2 (Figure 8) indicate that the two distributions combine more intense overland flow with lower erodibility areas, thus reducing the overall sediment production.

In Figure 9 we use the modelled SDR as a measure of sediment transfer connectivity, as it quantifies the proportion of mobilized sediment that is routed to the outlet of a selected subbasin by action of overland and channel flow. As such, the modelled SDR can be seen as a dynamic indicator of functional connectivity, where the discharge is represented explicitly in time and space as a function of the hydrological forcings and topographic characteristics, as opposed to the widely used approximation as a function of the upstream area. In this way, our approach integrates the variability of functional connectivity

both in time and space. A comparable approach to dynamically quantify functional connectivity has been proposed by Mahoney et al. (2018), which is also based on hydrological modelling.

The sediment delivery ratio shows that the connectivity of sediment sources is reduced by the spatial variability of precipitation and this effect can be explained by the geomorphic connectivity of the catchment. Higher precipitation, shallower soils, and steeper slopes in the southeastern region of the basin, i.e. tributaries T1, T3, T6 and the upper stretch of the main channel (see

Figure 9a), favor overland flow generation, and thus hydrological connectivity. However, the lower topographic connectivity of these subbasins overall causes a reduction in the sediment transfer connectivity. Such lower connectivity is indicated by the low SDRs of these subbasins in SIM 3, which does not account for the land use effect, and suggests the presence of geomorphic sediment buffers (Fryirs, 2013). The different topographic connectivity of the southeastern and northwestern regions reflects the different geomorphology of these two parts of the basin. In fact, the southeastern region of the basin is characterized by a

predominantly Last Glacial Maximum landscape with wide valleys and major instabilities, which are in most cases not directly connected to the river network (Schwab et al., 2008; Van Den Berg et al., 2012; Clapuyt et al., 2019). On the other hand, the northwestern part of the basin, i.e. tributaries T4 and T5, shows a rejuvenating landscape where recent fluvial dissection created narrow and deeply incised valleys with a strong coupling between hillslopes and channels (Schlunegger and Schneider, 2005; Norton et al., 2008).

The reduction of sediment transfer connectivity by spatially distributed surface erodibility can be attributed to the assumption in the sediment module that the sediment discharge always satisfies the overland flow transport capacity. Based on this assumption, a spatially variable $\alpha$ allows, on the one hand to modulate the sediment mobilization in space and, on the other hand, to define preferential areas of sediment deposition and therefore sediment connectivity. By associating a lower transport capacity to forests, their role as sediment buffers blocking sediments will emerge. Vice versa, high $\alpha$ values in crop areas will

mean the absence of obstacles to sediment flux. Therefore, the smaller sediment transfer connectivity of SIM 1 and 2 compared to SIM 3 and 4 reflects the location of sediment buffers (i.e. forests) with respect to the channel network. In fact, in most of the basin, forested areas surround channel headwaters, thus disconnecting the sediment sources on the hillslopes and mountain flanks from the river network (see e.g. Clinnick, 1985; Parkyn et al., 2005; Schoonover et al., 2006; Mekonnen et al., 2015).

**5.3 Sediment load and connectivity**

The analyses presented in the previous sections focus on the driving processes of sediment mobilization and transfer across the basin and the reasons for the reduction in SDR with variable erosion drivers. In this section we analyse how their balance determines the sediment load at the outlet.

In the distributed surface erodibility simulations (SIM 1 and 2) a reduced sediment yield (Y) is observed at the basin outlet determined by a reduction in both sediment production (P) and sediment transfer connectivity (expressed by the SDR) with

415 respect to uniform erodibility simulations (SIM 3 and 4):

$$\downarrow Y = SDR \downarrow \cdot P \downarrow .$$
(6)

In the distributed precipitation simulations (SIM 1 and 3) instead, an increased sediment yield at the basin outlet is observed compared to uniform precipitation simulations, which results from a combination of a smaller SDR and a much greater sediment production across the basin. The increase in sediment yield indicates that the greater sediment production dominates over the decreased sediment connectivity:

$$\uparrow Y = SDR \downarrow \cdot P \Uparrow . \tag{7}$$

This result means that localized sediment source areas are activated by the very high erosive power of localized precipitation captured by distributed simulations. Their signal reaches the outlet despite the system being globally less efficient in evacuating the eroded sediments. These hotspots of erosion are generated where precipitation falls with a high intensity, soil saturation is reached soon during storms, eventually favoured by shallow soils, and therefore hydrological and sediment flux connectivity are locally high.

In a hydrological modeling experiment conducted with TOPKAPI-ETH on the same catchment, Paschalis et al. (2014) demonstrated the dependence of the discharge peak on the clustering of high soil moisture areas. Our results show that the high soil moisture areas may also define the sediment signal. This finding also suggests that a large proportion of the sediment yield can be supplied by just few localized sediment sources (e.g. Pelletier, 2012). The role of soil moisture in producing high sediment concentrations has also been highlighted by Dominic et al. (2015) and Brasington and Richards (2000), who attribute the peaks of SSCs to the connection of remote sediment sources during the wetting up of the catchment.

Given the relevance of soil moisture spatial distribution for runoff generation, we also expect event sediment yields to be more affected by precipitation spatial variability, i.e. precipitation intensity, at low initial soil moisture than at high initial soil moisture, as is suggested by Figure 10b. This is further supported by findings of Paschalis et al. (2014) and Shah et al. (1996) which indicate that higher initial basin saturation reduces the dependency of runoff on precipitation spatial distribution. However, we also stress that in our study the relatively small difference between the sediment load distributions of low and high $SM_0$ events and the tendency to overestimate flow in low $SM_0$ events, do not allow for a clear conclusion.

## 6  Conclusions

We presented a new spatially distributed soil erosion and suspended sediment transport module integrated into the computationally efficient physically based hydrological model TOPKAPI-ETH. The model allows for continuous long-term, high temporal and spatial resolution simulations of erosion and sediment transport in mesoscale basins, and it is based on the physically driven processes of overland flow on hillslope and in channels. With the aim of exploring the impacts of two key spatially variable erosion drivers on suspended sediment dynamics, we conducted a series of numerical experiments on a mesoscale river basin. We compared the effects of spatially variable rainfall and surface erodibility with combinations of uniform and variable spatial distributions of these drivers.

Our results show that, first, the proposed model can reproduce part of the scatter of the observed SSC-Q relation, which is generated by spatially and temporally variable meteorological inputs and spatial heterogeneities of the physical properties

of the basin, leading to a multitude of possible flow and sediment pathways. At the same time, our results suggest that other processes are also relevant to capture the scatter, such as localized sediment sources and the inherent randomness of sediment production and transfer, which are not included in our model.

Second, we found that spatial variability in both drivers favors the clustering of sediment source areas and reduces their overall connectivity to the river network, by capturing the buffering effect of forests and low slope areas. At the same time, spatially variable surface erodibility reduces sediment production, while a spatially variable precipitation increases sediment production by high rates of erosion in areas of high rainfall and overland flow intensity.

Third, we found that the combination of the effects of spatial variability on sediment production and connectivity determines an overall lower sediment yield for distributed surface erodibility, due to reduced sediment production and to buffering effects, and a greater sediment yield for distributed precipitation, due to locally very high soil erosion. This last result is due to areas of high soil moisture in the catchment that are easy to saturate, which produce high local sediment inputs and catchment loads in spatially variable simulations.

Although our findings were obtained with reference to the specific climatic and geomorphologic features of the Kleine Emme catchment, we think they indicate the general importance of resolving the spatial variability in sediment mobilization and transfer processes when modelling sediment dynamics at the basin scale. The model we presented is particularly suitable for applications at medium and large scales, where gradients in climatic and physiographic characteristics represent a key control on sediment mobilization and transfer. Moreover, this model offers a valuable tool for investigating future scenarios of precipitation and land cover, which are expected to take place due to climate change or human land use management.

*Data availability.* DEM, soil and land use maps, discharge and suspended sediment concentrations data and simulation results are available at https://doi.org/10.3929/ethz-b-000358874. Meteorological input data can be requested at https://gate.meteoswiss.ch/idaweb/login.do.

*Author contributions.* GB developed the model, carried out the simulations and the analyses of the results. PM and PB contributed to the conceptualization of the model and to the discussion of the results. GB prepared the manuscript with contributions and edits from all co-authors.

*Competing interests.* The authors declare that they have no conflict of interest.

*Acknowledgements.* The study was funded by the DAFNE project, an Horizon 2020 programme WATER 2015 of the European Union, GA no. 690268. We thank Fritz Schlunegger (University of Bern) for sharing his knowledge of the study basin, Enrico Weber (ETH Zurich) and Scott Sinclair (ETH Zurich) for technical support during the development of the model.

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
