# Peer review of "In the following, additional information on the model inputs and the calibration process are reported."

_Earth Surface Dynamics, 2019_

## Referee Comment (RC1) · Anonymous Referee #1 · 13 Jan 2020

The manuscript describes a new suspended sediment flux model which is then used to analyze sediment dynamics and sources in a mid-size catchment. The paper is well written. Unfortunately, I have great reservations about the model novelty and the interpretation of the results. I recommend Major Revisions as I think that the manuscript can be of interest once it is more properly framed.

General comments: The authors all but ignored existing large-scale sediment flux models (see a most relevant review paper and a couple of examples below). This is a major emission that must be corrected; their model should be framed in reference to these models.

The authors greatly over-sell the novelty and capabilities of the sediment model. While it is true that the hydrological framework is physically-based, the sediment model is a

simple empirical equation (Eq. 2) that predicts sediment as a function of discharge, slope and a spatially variable (alpha) coefficient. Alpha is calibrated using USLE parameter combination. Sediment transport (Eq. 4) is a simple cell-to-cell and time-step balance. I see very little novelty in this model. The authors must make the argument of why this model is novel if they wish to continue claiming it is (this is a cornerstone of the manuscript at the moment).

The evaluation of the sediment model is odd - referring to the relatively low scatter in the SSC-Q plot (Fig 3) as an argument for strong model performance. A standard model performance analysis is offered for the model's hydrological predictions (Table 1). It seems that the observed sediment is used for model calibration so we actually left with little knowledge about how well the model is doing.

Given the relative simplicity of the model and the way it was calibrated, the interpretation of the model results extends much beyond the model's ability to represent the discussed processes. The authors need to frame their analysis within the model's capabilities to represent the relevant processes and drivers. Some examples of overreaching are 1st sentence in the Discussion, sentence starting in lines 300, 311 & 315.

Example of references to consider: A review paper which evaluates 14 models: De Vente, J., Poesen, J., Verstraeten, G., Govers, G., Vanmaercke, M., Van Rompaey, A., ... & Boix-Fayos, C. (2013). Predicting soil erosion and sediment yield at regional scales: where do we stand?. Earth-Science Reviews, 127, 16-29.

Cohen, S., A.J. Kettner, J.P.M. Syvitski, B.M. Fekete (2013). WBMsed, a distributed global scale riverine sediment flux model: model description and validation Computers & Geosciences, 53, pp. 80-93

[cited in the paper but not in reference to the model] Pelletier, (2012). A spatially distributed model for the long-term suspended sediment discharge and delivery ratio of drainage basins. J. Geophys. Res., 117 (F2), p. F02028

Syvitski, J.D. Milliman (2007). Geology, geography, and humans battle for dominance over the delivery of fluvial sediment to the coastal ocean. J. Geol., 115 (1) (2007), pp. 1-19

---

## Short Comment (SC1) · 17 Jan 2020

The authors investigate the contributions of spatial variability of precipitation and soil erosion parameters on the variability of suspended sediment transport using a distributed model of hydrology, hillslope erosion, and suspended sediment transport. The research is very well designed, implemented and written. The derivation and calibration of the surface erosion parameters was novel. I have no comments on the presentation of this research and the text. I have a few questions for the authors to consider in revising this manuscript.

I was wondering how the SIMs 1, 2, 3, 4 results would look like when plotted as a suspended sediment rating curve as in Fig 3b in comparison to observations?

[Figure]

Similarly, I would suggest adding observed sediment variability to Fig 9a as box-whisker plots.

I was expecting to see bigger SDR values, based on the description of the channels having bedrock exposure and the model not allowing any exchange of suspended sediment with the bed, E=0. With these low SDRs there should be deposition in the channel and if E=0 how is deposition is modeled? And even if the model allows deposition but not re-suspension how can that assumption be justified. Some clarification on this would be appreciated by readers.

Unless I missed this in the paper, I was wondering how do variability of precip and soil parameters contribute to the observed variability of sediment quantitatively. Could this sediment flux variability be quantified in terms of the variability of precip and soil erosion in an expression. Of course discussions of this nature can go all the way to information content, entropy and so on, but some discussion of whether more/less information in model parameters and forcing can be attributable to the changes in the variability of observed/model sediment would be interesting to see.

erkan istanbulluoglu

---

## Author Comment (AC1) · 25 Jan 2020

**Response to Referee 1**

1. Model Context. The referee states that "*The authors all but ignored existing large-scale sediment flux models (see a most relevant review paper and a couple of examples below). This is a major emission that must be corrected; their model should be framed in reference to these models.*" In the submitted manuscript we indeed referenced only those models that in our opinion were directly comparable to the approach we took, like tRIBS (Francipane et al., 2012) and the model of Tsuruta et al. (2018), see lines 62-69. But, as the reviewer suggests, we will add a brief review of other selected physics-based large-scale approaches in the revised manuscript to help frame our work. We will highlight the differences between the approaches (see Section 2 below). At the same time, we prefer to stay focused in the paper on physics-based modelling approaches only. Most of the models reviewed or presented in the papers suggested by the referee (De Vente et al., 2013; Cohen et al., 2013; Pellettier, 2012; Syvistky and Milliman, 2007) are statistical and steady state models which cannot be seen as reference models, because they are developed for a different purpose and cannot answer the same questions we address in our work.

2. Novelty. The referee states that "*The authors greatly over-sell the novelty and capabilities of the sediment model. While it is true that the hydrological framework is physically-based, the sediment model is a simple empirical equation (Eq. 2) that predicts sediment as a function of discharge, slope and a spatially variable (alpha) coefficient… Sediment transport (Eq. 4) is a simple cell-to-cell and time-step balance. I see very little novelty in this model. The authors must make the argument of why this model is novel if they wish to continue claiming it.*" This point and criticism will require deeper explanation in the revised manuscript because we have clearly not managed to get the message through. Indeed the novelty is not in the sediment model per se, but in the combined hydrology-sediment system approach and the questions it allows to address.

The novelty we perceive is based on the combination of the following elements: (a) We combine physically-based unsteady hydrological simulation of surface overland flow with a simple hillslope erosion and sediment transport component. This ensures that sediment is produced and transported along hillslopes by overland flow respecting physical processes of hillslope erosion and sediment transport as we understand them. The sediment component is simple by design (sediment production and continuity in Eqs 2 and 4 mentioned above), so that the most uncertain part of the modelling system is not over-parameterized. (b) The high spatial and temporal resolutions of the model (100 m and 1 hr) allow the inclusion of detailed topographic variations, connectivity of sediment pathways in space and time, and fast response to heavy precipitation where it happens. (c) Continuous simulation (order of decades) by our approach, allows to track overland runoff generation and hillslope sediment transport by spatially distributed changes in soil moisture, snowmelt, and rainfall, not only for individual events, but over long periods of time reflecting also long-term changes in soil moisture states, rainfall seasonality, etc.

None of the physics-based models reviewed in De Vente et al. (2013) or mentioned in the introduction of the manuscript (lines 62-69) combines these three characteristics at a spatial scale comparable to our case study. This is the context in which we perceive the novelty of our work, and which allows us to explore the effects of the spatial variability in catchment erodibility and rainfall with higher confidence.

At the same time, we recognize that our model is not novel in the sense that it is the first and only such model. For example, it is similar to tRIBS (Francipane et al., 2012) and the model of Tsuruta et al. (2018). However, the former is not applicable to large catchments and long simulations at high resolutions due to computational demand, while our setup is computationally very efficient and applicable to medium and large-scale basins, and the latter is a coarser resolution model with less

physical hillslope surface runoff generation routines. More details about the spatial and temporal resolutions of these models and the physics-based approaches reviewed in De Vente et al. (2013) are summarized in Appendix A.

In conclusion, we do think that our approach has unique strengths that allow us to explore the hydrology-sedimentology connections leading to sediment generation pathways at high resolutions, which other approaches do not have. In the revised manuscript we will carefully review the text to make sure this aim and the context of the work with respect to existing models is clear.

3. Model Evaluation. The referee states that "*The evaluation of the sediment model is odd - referring to the relatively low scatter in the SSC-Q plot (Fig 3) as an argument for strong model performance. A standard model performance analysis is offered for the model's hydrological predictions (Table 1). It seems that the observed sediment is used for model calibration so we actually left with little knowledge about how well the model is doing. Given the relative simplicity of the model and the way it was calibrated, the interpretation of the model results extends much beyond the model's ability to represent the discussed processes…*". Indeed the referee is correct that it is much easier to calibrate the hydrological part of the model than the sedimentological one, mainly because we do not have the data to do so. There is only one suspended sediment measurement point at the outlet of the basin where bi-weekly measurements are available for a reasonably long period. We do not consider it meaningful to tweak the simple advection-based sediment transport routine implemented in the model, to match "perfectly" the observed hourly concentrations at the outlet measured twice a week. A comparison of sediment transport at the daily scale is also not possible with the given resolution of the measurements. Rather we assumed as a qualitative measure of success the reproduction of properties of the observed sediment rating curve (SRC), i.e. the relationship between hourly discharge and suspended sediment concentration (SSC), which captures the catchment sediment dynamics. Concretely, we calibrate the sediment model parameters, i.e. the river initiation threshold RT and the $\alpha_1$ erodibility parameter (Eq. 5), to (a) reproduce the observed slope of the SRC as best as we can, as well as (b) the frequency distribution of observed SSCs. As a quantitative indicator of the model performance, we propose to introduce in the revised manuscript the percentage of modelled SSCs that fall within the 5th and 95th percentile of the observations and this equals to 90.4% in our simulation. We also omitted a traditional validation with a part of the dataset not used in calibration as our observed records are too short and our main focus is on the sensitivity to input data (spatial variability in surface erodibility and rainfall) not on the predictive uncertainty in SSC per se.

Regarding the last point that the "*…interpretation of the model results extends much beyond the model's ability to represent the discussed processes…*" we do not fully agree with the referee. The spread around the SRC in our deterministic approach is due to (a) the spatially distributed nature of the model, which allows to simulate the heterogeneous response of the basin to hydrological forcing, based on the topographic characteristics, depth and properties of the soil, (b) the spatial variability of surface erodibility and the connectivity of hillslope flow paths to the river network, and (c) the spatio-temporal distribution of rainfall leading to overland flow and erosion (lines 293-302). We are of the opinion that all of these processes are robustly included in our modelling approach. They of course cannot explain all the SRC spread because in the real natural catchments there is an added element of stochasticity in sediment mobilization, transport, and sediment supply limitations, which add to the SRC variability (lines 306-312). However, we believe that this does not invalidate our modelling results or their ability to provide insights into process effects, like the role of spatial variability in erosion drivers. In the revision we will be more clear on these limitations of the results and their interpretations.

**References**

Cohen, S., A.J. Kettner, J.P.M. Syvitski, B.M. Fekete (2013). WBMsed, a distributed global scale riverine sediment flux model: model description and validation Computers & Geosciences, 53, pp. 80-93.

De Vente, J., Poesen, J., Verstraeten, G., Govers, G., Vanmaercke, M., Van Rompaey, A., ... & Boix-Fayos, C. (2013). Predicting soil erosion and sediment yield at regional scales: where do we stand? Earth-Science Reviews, 127, 16-29.

Francipane, A., Ivanov, V. Y., Noto, L. V., Istanbulluoglu, E., Arnone, E., and Bras, R. L. (2012). TRIBS-Erosion: A parsimonious physically-based model for studying catchment hydro-geomorphic response, Catena, 92, 216–231, https://doi.org/10.1016/j.catena.2011.10.005.

Pelletier (2012). A spatially distributed model for the long-term suspended sediment discharge and delivery ratio of drainage basins. J. Geophys. Res., 117 (F2), p. F02028.

Syvitski, J.D. Milliman (2007). Geology, geography, and humans battle for dominance over the delivery of fluvial sediment to the coastal ocean. J. Geol., 115 (1) (2007), pp. 1-19.

Tsuruta, K., Hassan, M. A., Donner, S. D., and Alila, Y. (2018). Development and Application of a Large-Scale, Physically Based, Distributed Suspended Sediment Transport Model on the Fraser River Basin, British Columbia, Canada, Journal of Geophysical Research, 123, 2481–2508, https://doi.org/10.1029/2017JF004578.

**Appendix A:**

Comparison of spatial and temporal scales from "Physics-based Models" in DeVente et al. (2013), tRIBS, Tsuruta et al. (2018) and the model presented in the manuscript.

|  | SPATIAL SCALE AND RESOLUTION | TEMPORAL SCALE AND RESOLUTION |
|---|---|---|
| **AGNPS** [1,2] | Basin scale: <2.3 km$^2$
Discretization: sub-basins | Continuous
Daily |
| **LISEM** [3,4,5] | Basin scale: <5.7 km2
Resolution: 10/20 m | Individual rainfall events
Minutes |
| **PESERA** [6,7,8] | European scale
Resolution: 1 km | Steady state |
| **SWAT** [9,10] | Basin scale: up to 185 000 km$^2$
HRUs (resolution 1-100 km$^2$) | Continuous
Daily |
| **WBMsed** [11] | Global scale
Resolution: ~11-55 km | Steady state
Representative daily sediment flux |
| **Pellettier, 2012** | Global scale
Resolution: 10 km | Steady state |
| **tRIBS** [12] | Basin scale: 0.037 km$^2$
Multiple resolution (irregular mesh) | Continuous
Minutes/hours |
| **Tsuruta et al., 2018** | Basin scale: 230 000 km$^2$
Resolution: ~7 km | Continuous
Hourly |
| **This paper** | Basin scale: 477 km$^2$
Resolution: 100 m | Continuous
Hourly |

[1] Licciardello, F., Zema, D.A., Zimbone, S.M., Bingner, R.L., 2007. Runoff and soil erosion evaluation by the AnnAGNPS model in a small Mediterranean watershed. Trans. Am. Soc. Agric. Biol. Eng. 50 (5), 1585–1593.

[2] Haregeweyn, N., Yohannes, F., 2003. Testing and evaluation of the agricultural non-point source pollutionmodel (AGNPS) on Augucho catchment, western Hararghe, Ethiopia. Agric. Ecosyst. Environ. 99 (1–3), 201–212.

[3] Jetten, V., de Roo, A., Favis-Mortlock, D., 1999. Evaluation of field-scale and catchment-scale soil erosion models. Catena 37 (3–4), 521–541.

[4] Hessel, R., van den Bosch, R., Vigiak, O., 2006. Evaluation of the LISEM soil erosion model in two catchments in the East African Highlands. Earth Surf. Process. Landforms 31 (4), 469–486.

[5] Takken, I., Beuselinck, L., Nachtergaele, J., Govers, G., Poesen, J., Degraer, G., 1999. Spatial evaluation of a physically-based distributed erosion model (LISEM). Catena 37 (3–4), 431–447

[6] Kirkby,M.J., Irvine, B.J., Jones, R.J.A., Govers, G., 2008. The PESERA coarse scale erosion model for Europe. I. Model rationale and implementation. Eur. J. Soil Sci. 59 (6), 1293–1306

[7] de Vente, J., Poesen, J., Verstraeten, G., Van Rompaey, A., Govers, G., 2008. Spatially distributed modelling of soil erosion and sediment yield at regional scales in Spain. Glob. Planet. Chang. 60 (3–4), 393–415.

[8] Van Rompaey, A.J.J., Vieillefont, V., Jones, R.J.A., Montanarella, L., Verstraeten, G., Bazzoffi, P., Dostal, T., Krasa, J., de Vente, J., Poesen, J., 2003. Validation of soil erosion estimates at European scale. European Soil Bureau Research Report No.13, p. 26.

[9] Tuppad, P., Kannan, N., Srinivasan, R., Rossi, C., Arnold, J., 2010. Simulation of agricultural management alternatives for watershed protection. Water Resour. Manage. 24 (12), 3115–3144.

[10] Betrie, G.D., Mohamed, Y.A., van Griensven, A., Srinivasan, R., 2011. Sediment management modelling in the Blue Nile Basin using SWAT model. Hydrol. Earth Syst. Sci. 15 (3), 807–818

[11] Cohen, S., Kettner, A.J., Syvitski, J.P.M., Fekete, B.M., 2013. WBMsed, a distributed globalscale riverine sediment flux model: model description and validation. Computers & Geosciences 53, 80–93

[12] Francipane, A., Ivanov, V. Y., Noto, L. V., Istanbulluoglu, E., Arnone, E., and Bras, R. L. (2012). TRIBS-Erosion: A parsimonious physically-based model for studying catchment hydro-geomorphic response, Catena, 92, 216–231, https://doi.org/10.1016/j.catena.2011.10.005.

---

## Author Comment (AC2) · 28 Jan 2020

**Response to Interactive Comment by Erkan Istanbulluoglu**

In the following, we report the text of the Interactive Comment in blue italic, and in black our reply.

*The authors investigate the contributions of spatial variability of precipitation and soil erosion parameters on the variability of suspended sediment transport using a distributed model of hydrology, hillslope erosion, and suspended sediment transport. The research is very well designed, implemented and written. The derivation and calibration of the surface erosion parameters was novel. I have no comments on the presentation of this research and the text. I have a few questions for the authors to consider in revising this manuscript.*

We thank Erkan Instanbulluoglu for the interest in our work and for the constructive questions asked. The answers to the points 1 and 3 below are based on analyses that were not included in the manuscript. Given the interest, we will consider adding these figures and discussion points to the supplementary materials of the revised manuscript.

*1. I was wondering how the SIMs 1, 2, 3, 4 results would look like when plotted as a suspended sediment rating curve as in Fig 3b in comparison to observations? Similarly, I would suggest adding observed sediment variability to Fig 9a as box-whisker plots.*

[Figure]

Figure 1: Density plot of simulated SSC for SIM1 to SIM4 (left to right) sampled at the time of measurements, compared with measurements (lines give median and 15-85 percentiles).

Figure 1 compares the modelled SSC (density plots) in SIM 1 to 4 with the observations (lines). The comparison of SIM 1 and 3 with SIM 2 and 4 shows the effect of the spatial distribution of precipitation in stretching the bulk of the modelled concentrations towards higher values, which reflects the increase in the annual sediment load. Analogously, the effect of spatial distribution of $\alpha$ is opposite (compare SIM 1 and 2 with SIM 3 and 4). The plots are in log-log scale, so we point out that the differences between the simulations are more relevant at high concentrations.

In Figure 2 (modification of Fig 9a of the manuscript) the simulated annual sediment load variability is compared with the observed variability. The observed sediment loads have been computed by fitting sediment rating curves (SSC=aQ$^b$) to the observations and by using them to estimate the SSC corresponding to each observed hourly discharge in the simulation period. Two estimates of annual observed sediment loads are proposed based on the available measurements. The loads in "OBS A" have been computed based on a single sediment rating curve fitted to the 13 years of SSC observations, while "OBS B" loads have been computed by fitting a rating curve to each year of SSC observations, with the aim of better representing the interannual variability of the sediment load. We observe that the estimated mean annual loads are lower than 2.83 10$^5$ t/y (line 233), proposed by Hinderer et al., (2013) and derived from BAFU (2010). The reason is that the latter is not based on sediment rating curves, but on an estimate of the mean daily load derived from the observed Q-SSC pairs of points that gives more weight to the high observed SSCs.

The figure also shows that the simulations only capture a fraction of the total observed interannual variability. The reason for this underestimation is that observed variability in SSCs also reflects the activation of local sources of sediment heterogeneously distributed across the basin, the stochasticity in mobilizing and transporting sediment on the hillslope pathways and local sediment

supply limitations, which may not be represented by the soil depth as assumed in the model (lines 306-312).

[Figure]

*Figure 2: Boxplots of simulated and observed annual sediment loads. "OBS A" loads are computed with a unique SSC-Q rating curve fitted to the whole observation period, "OBS B" with yearly sediment rating curves.*

*2. I was expecting to see bigger SDR values, based on the description of the channels having bedrock exposure and the model not allowing any exchange of suspended sediment with the bed, E=0. With these low SDRs there should be deposition in the channel and if E=0 how is deposition is modeled? And even if the model allows deposition but not re-suspension how can that assumption be justified. Some clarification on this would be appreciated by readers.*

The model does not allow deposition and re-suspension of fine material carried in suspension in the channels. The low SDRs are explained by the deposition of mobilized fine sediments on the hillslopes themselves, before they reach the channel. Given that hillslope sediment production (erosion) is assumed to satisfy transport capacity (equation 2 in the paper) then sediment discharge and local deposition is driven by changes in slope, overland flow discharge, or land cover. Deposition is particularly strong in the areas of low hillslope-channel connectivity, such as the area upstream of point MC1 and in tributaries T1, T3, T6 (see lines 323 to 334). For catchments of the size of the Kleine Emme, annual SDR below 10% for fine sediment are not unexpected (Julien, 1995).

*3. Unless I missed this in the paper, I was wondering how do variability of precip and soil parameters contribute to the observed variability of sediment quantitatively. Could this sediment flux variability be quantified in terms of the variability of precip and soil erosion in an expression. Of course discussions of this nature can go all the way to information content, entropy and so on, but some discussion of whether more/less information in model parameters and forcing can be attributable to the changes in the variability of observed/model sediment would be interesting to see.*

It is not possible to quantify the variability in sediment flux as a function of precipitation and surface erodibility variability analytically (exactly), if that is what was meant by the question. But we can provide some insights from the numerical simulations. For example, we can compare the scatter of the simulated SSC scenarios against each other and against observations.

To quantify the scatter of the SSC-Q points in Figure 1 independently of the mean simulated SSC, we binned the simulated discharges, computed the coefficients of variation (CVs) of the sediment concentrations in each discharge bin and reported them as a boxplot for all discharges in Figure 3. We observe that the distribution of the CVs shifts to lower values every time a source of variability (rainfall or $\alpha$ distribution) is removed, therefore, we observe a general correspondence between information content of the inputs and scatter of the predictions of SSC.
However, we also observe that the changes between simulations are very small, especially in the mean value, thus suggesting that the spatially distributed nature of the model itself plays a more relevant role than the variability of the analysed input variables (rainfall and surface erodibility).

The comparison of observed and simulated CVs, shows the amount of variability of the lower 85[th] percentile of observed SSCs that is captured by the model. As expected, the observed variability is much larger than the simulated one, because of the sources of variability which are not accounted for in our model (see point 1).

[Figure]

Figure 3: Boxplots of the coefficients of variation of the SSC-Q relation for the four simulations (left), and comparison with observed SSC smaller than the 85[th] percentile (right).

**References**

Hinderer, M., Kastowski, M., Kamelger, A., Bartolini, C., and Schlunegger, F.: River loads and modern denudation of the Alps - A review, Earth-Science Reviews, 118, 11–44, https://doi.org/10.1016/j.earscirev.2013.01.001, 2013.

BAFU. Hydrologisches Jahrbuch der Schweiz, 2010, Bundesamt für Umwelt (BAFU), https://www.bafu.admin.ch/bafu/de/home/themen/wasser/zustand/daten/hydrologisches-jahrbuch.html, 2010

Julien, P.Y.: Erosion and sedimentation, Cambridge university press, 1995.

---

## Referee Comment (RC2) · Anonymous Referee #2 · 19 Feb 2020

The paper presents a hydrosedimentary model that couples the TOPKAPI-ETH hydrological model to a physically based and spatially distributed erosion and sediment transport model. Erosion on hillslopes is represented by the process of detachment/deposition by runoff using an empirical formulation. Sediment transport is provided by a mass conservation equation. The study examines the effect of spatial variability of precipitation and surface erodibility on sediment dynamics by performing continuous simulations over a period from 2003 to 2016, the first year being considered as a warm-up period for the model. In the simulation of reference, precipitation and surface erodibility are spatially distributed. In the other three scenarios, either precipitation, surface erodibility or both variables are spatially averaged over the catchment area. The results are analyzed using the Q-SSC relations at the catchment outlet, the erosion-deposition maps at the end of the simulated period and the terms of gross erosion, sediment yield and sediment delivery ratio. The topic covered by the article is original and of interest to the journal. The scale of the study (mesoscale catchment) is little discussed in the literature and the fact that there are two control points within the catchment in addition to the outlet is an added value to be highlighted. The article is well written. The authors provide the data used in the article in the « data availability » section at the end of the paper.

However, the questions asked are not precise enough. It is a bit ambitious to want to answer such generic questions with only 4 scenarios. The model is very little evaluated in terms of erosion before analysing the results of the different scenarios. It is therefore difficult to give credit to the results obtained. My advice would be to reformulate questions thar are compatible with the framework offered by the tested scenarios and to rework the results and discussion sections according to these new questions.

I therefore recommend major revisions for this paper.

General remarks :

- The authors do not mention the DHSVM model although it would be a very relevant tool for this type of catchment. It is necessary to justify the development of a new model compared to existing models such as DHSVM.

- The description of the erosion model did not seem clear enough to me, especially the distinction between the representation of hillslopes and river processes.

- The authors use data from Swiss operational services. However the temporal frequency of SSC data is too low for a catchment of this size located in a mountainous area. Flood events are most likely under-sampled. High SSC values are probably missing from the data set for this reason.

- One could be interested in the impact of the scenarios on the hydrological response and the indirect impact this may have on sediment dynamics.

- The concepts of structural and functional connectivity, widely present in the literature, are not discussed although they are at the heart of the subject developed in the paper.

- Connectivity indices are not used.

- The process of detachment by rain is not taken into account in the model. Only the process of detachment by runoff is taken into account. This is questionable when the objective is to estimate the effect of spatial variability of precipitation.

- Connectivity index maps could be used to study the spatial organization of erosion (Section 4.2). It is questionable whether there is any real added value in using the model presented in this study to address this issue.

- In section 3.4, the authors examine the results at the temporal scale of the flood event. It is difficult to examine the effect of soil moisture on erosion and sediment transport without giving guarantees on the performance of the model in reproducing flows under dry and wet conditions.

- The summary at the beginning of Section 4.3 is interesting.

Specific remarks :

-p2 l33: I would suggest adding "especially in small to medium catchments (up to 1000 km2) after "the strong non-uniqueness of suspended sediment concentrations (SSCs)".

-p2 l37: I would suggest adding "and transfers " after "in sediment mobilization".

-p2 l39 to 52: rewrite this part which is not clear and take into account the concepts of structural and functional connectivity.

-p2 l48: add reference Misset et al (2018)

Misset C., Recking A., Legout C., Poirel A., Cazilhac M., Esteves Michel, Bertrand M. (2019). An attempt to link suspended load hysteresis patterns and sediment sources configuration in alpine catchments. Journal of Hydrology, 576, 72-84. ISSN 0022-1694

- p2 l56: replace "transport" by "transfer" in several places in the text

- p3 l65: replace "cesar-lisflood" with "caesar-lisflood".

-p3 l62 to 75: add the reference to the DHSVM model and explain the added value of the model presented in relation to this model

-p3 l77: "a physically explicit spatially distributed deterministic model": simplify the formula. What does "explicit" mean here?

- p3 l83: "mean annual discharge" instead of "average discharge".

- p3 l87 : " mostly driven by overland flow ". What about rainfall processes ?

- p4 l97: what is the scale for the soil map?

- p4 l103: Is it really 2D whereas the equations presented p5 are 1D?

- p5 l105 to 114: I do not understand how the hydrographic network is represented and discretized. The same question applies to the hillslopes. A specific part is missing for describing the discretization used  in the model.

- p5 l107: " catchment scale ": it is not precise enough. What scale?

- p5: put the dimensions of the variables presented in the equations. I do not understand the distinction between hillslopes and rivers in terms of erosion and transport processes. What is the link between the terms D and E?

- p5 Eq.4: I do not understand the definition of X. It should be a width rather than a length for the calculation of the flux.

- p7 l166-167: is this a wash load hypothesis?

- p8 Fig3 : SSC values seem low for a mountainous catchment area. This is certainly related to the lack of observed values during floods.

- p8 l186-188: it is questionable to use the slope of the Q-SSC relation given the dispersion that exists between these two variables (even in log scale)

- p10 l226-228: what forms of erosion are observed within the basin?

- p10 l234: "Hinderer et al. (2013)" is not present in the reference list.

- p10 l237: "The underestimation of sediment load (...) we do not like to reproduce the largest measured sediment concentrations". This is a working hypothesis that should be placed in « Material and Method ».

- p10 Fig5: indicate the observed data as red dots on the SSC time series.

- p11 Fig6(a): over which periods are the intensities calculated: over the rain periods only or over the whole period of simulation?

- p11 l242: I suggest modifying "where SIM 2 and 3 are compared respectively with SM1".

- p13 Fig8(b): There is a black dot without a text caption

- p17 l369-372: I am not convinced by this hypothesis, which depends heavily on the nature of the soils and the infiltration model used.

---

## Author Comment (AC3) · 5 Mar 2020

**Response to Referee 2**

In the following, we report the text of the review in blue italic, and in black our reply.

*The paper presents a hydrosedimentary model that couples the TOPKAPI-ETH hydrological model to a physically based and spatially distributed erosion and sediment transport model. […] The authors provide the data used in the article in the « data availability » section at the end of the paper.*

*However, the questions asked are not precise enough. It is a bit ambitious to want to answer such generic questions with only 4 scenarios. The model is very little evaluated in terms of erosion before analysing the results of the different scenarios. It is therefore difficult to give credit to the results obtained. My advice would be to reformulate questions thar are compatible with the framework offered by the tested scenarios and to rework the results and discussion sections according to these new questions.*
*[…]*

We will modify the research questions in order to be more specific on the expectations of the case study and the simulations that we performed. Results and conclusions will be adapted accordingly. We will also support some of the discussion statements with further analyses of the hydrological results, as suggested in some of the comments below.

*General remarks:*

- *The authors do not mention the DHSVM model although it would be a very relevant tool for this type of catchment. It is necessary to justify the development of a new model compared to existing models such as DHSVM.*

(1) DHSVM is indeed a relevant tool in the framework of these type of models. DHSVM features a rigorous description of the hydrological processes and the role of vegetation and simulates sediment production by hillslope erosion, road erosion and mass wasting. TOPKAPI-ETH presents a slightly more simplified description of the hydrological processes and only includes erosion by overland flow. These choices are aimed at avoiding over-parameterization of the model and at keeping it computationally efficient and thus suitable for mesoscale catchment applications and small grid sizes, even when further components would be added (e.g. additional sediment transport processes).
We will introduce the DHSVM model in the literature review (Introduction) and highlight the differences with our approach.

- *The description of the erosion model did not seem clear enough to me, especially the distinction between the representation of hillslopes and river processes.*

(2) Sediment production and transport on the hillslopes is based on a transport capacity-mass balance approach, i.e. sediment flux is assumed to be always at transport capacity, and the model simulates erosion or deposition when there is a change in transport capacity. Sediment delivered to the channel network is advected by the river flow, there is no possibility for deposition or entrainment from the bed in the channel. The transition between sediment transport description as hillslope process to channel process corresponds to the transition from water flow routing as overland flow to channelized flow. This takes place between hillslope and channel cells and is fundamentally determined by the drainage area threshold (RT) used to identify river cells in the DEM, i.e. the river initiation threshold which was a parameter in our model

We will make sure that this concept is clearer in the revised manuscript, by reworking section 2.1.

- *The authors use data from Swiss operational services. However the temporal frequency of SSC data is too low for a catchment of this size located in a mountainous area. Flood events are most likely under-sampled. High SSC values are probably missing from the data set for this reason.*

(3)  We agree with the reviewer on this point and we will explicitly state this limitation of the data in the revised manuscript. We are aware that the observations are missing extreme SSCs during floods and, at the same time, the model underestimates highest SSCs coming from very localized sediment sources. For this reason in the calibration of the model we focused only on the lowest 85$^{th}$ percentile of the SSC dataset. There is nothing we can do about the temporal frequency of the data (twice a week). All sediment monitoring stations of the Federal Office of the Environment in Switzerland have such low frequency except for few automatic stations with turbidity measurements in recent years.

- *One could be interested in the impact of the scenarios on the hydrological response and the indirect impact this may have on sediment dynamics.*

(4)  This is a good point. We will add a comparison of the hydrological response in the 4 simulations, by evaluating the mean annual flow, the annual flood and the variability of flow at the basin outlet and the mean annual surface runoff on the hillslopes. This plot will be useful to support some of the discussion statements.

- *The concepts of structural and functional connectivity, widely present in the literature, are not discussed although they are at the heart of the subject developed in the paper.*

- *Connectivity indices are not used.*

  Both comments have been addressed in point (6) below.

- *The process of detachment by rain is not taken into account in the model. Only the process of detachment by runoff is taken into account. This is questionable when the objective is to estimate the effect of spatial variability of precipitation.*

(5)  Detachment by rain, together with the overland flow entrainment capacity, defines the amount of sediment available for transport. In our model, we do not simulate the local rainsplash detachment processes separately from the mobilization processes at the grid scale, rather we assume that sediment on the hillslopes is always available to fulfill the overland transport capacity. Sediment availability is only limited by the soil depth (sediment layer thickness), however, this limitation doesn't play a role in our simulations. It would be necessary to include the process of rainfall detachment by rainsplash if the model distinguished between the processes of sediment detachment, determining the sediment available for transport, and sediment mobilization.
  We agree that the manuscript is currently unclear in this regard and we will make sure this distinction is clearer in the revised version.

- *Connectivity index maps could be used to study the spatial organization of erosion (Section 4.2). It is questionable whether there is any real added value in using the model presented in this study to address this issue.*

(6) The vast majority of connectivity indices provides a static description of the structural or functional sediment connectivity, based on the upslope contributing area as a proxy for discharge to estimate stream power (Heckmann et al., 2018). The sediment delivery ratio SDR simulated by our model quantifies the proportion of eroded sediments that are routed to a point on a river network or outlet of a selected subbasin, by action of overland flow and channel flow. As such, SDR is a dynamic indicator of functional connectivity, where the discharge (and thus stream power) is considered explicitly as a function of the hydrological forcings and topographic characteristics, instead of being represented by the upstream area only. Besides accounting for the time dependency of discharge, SDR also integrates the variability in space of the functional connectivity, by substituting the unique Q-A relationship used in traditional connectivity indices, with the explicit simulation of overland flow on the hillslopes. We have applied this analysis at the subbasin scale, but SDR could be potentially computed for each grid cell to build a connectivity map. Mahoney et al. (2018) propose a comparable approach that quantifies dynamic functional connectivity, based on hydrological modelling too.
We believe that this is an interesting discussion point to complement section 4.2.

- *In section 3.4, the authors examine the results at the temporal scale of the flood event. It is difficult to examine the effect of soil moisture on erosion and sediment transport without giving guarantees on the performance of the model in reproducing flows under dry and wet conditions.*

(7) We will evaluate the hydrological model performance for the low and high initial soil moisture events analysed in section 3.4, by means of the performance indices used in Table 1. Accordingly, we will adapt the discussion of Fig 9 to take into account the new hydrological analysis. However, we do expect to see an effect of soil moisture on erosion and sediment transport, given the different hydrological response of the catchment that Paschalis et al. (2013) found for low and high initial soil moisture storms with the same hydrological model without sediment transport.

- *The summary at the beginning of Section 4.3 is interesting.*

*Specific remarks :*

*-p2 l33: I would suggest adding "especially in small to medium catchments (up to 1000 km2) after "the strong non-uniqueness of suspended sediment concentrations (SSCs)".*

*-p2 l37: I would suggest adding "and transfers " after "in sediment mobilization".*

*-p2 l39 to 52: rewrite this part which is not clear and take into account the concepts of structural and functional connectivity.*

*-p2 l48: add reference Misset et al (2018)*

*Misset C., Recking A., Legout C., Poirel A., Cazilhac M., Esteves Michel, Bertrand M. (2019). An attempt to link suspended load hysteresis patterns and sediment sources configuration in alpine catchments. Journal of Hydrology, 576, 72-84. ISSN 0022-1694*

*- p2 l56: replace "transport" by "transfer" in several places in the text*

*- p3 l65: replace "cesar-lisflood" with "caesar-lisflood".*

*-p3 l62 to 75: add the reference to the DHSVM model and explain the added value of the model presented in relation to this model*

See point (1)

*-p3 l77: "a physically explicit spatially distributed deterministic model": simplify the formula. What does "explicit" mean here?*

"Explicit" means that the model is based on a physical representation of most processes, but it still contains some conceptualisations or approximations of the processes. For general understanding, we will replace "explicit" with "based" in the revised manuscript.

*- p3 l83: "mean annual discharge" instead of "average discharge".*

*- p3 l87 : " mostly driven by overland flow ". What about rainfall processes ?*

This is correct, we will modify the sentence.

*- p4 l97: what is the scale for the soil map?*

The scale is given by the coordinates on the x- and y- axes (units are the same as in Fig 1a).

*- p4 l103: Is it really 2D whereas the equations presented p5 are 1D?*

The solution is 1D in the direction of the steepest descent at the grid scale for surface and subsurface flow. All inflows from the neighbourhood cells are integrated in space.

*- p5 l105 to 114: I do not understand how the hydrographic network is represented and discretized. The same question applies to the hillslopes. A specific part is missing for describing the discretization used in the model.*

The entire basin is discretized as a 100 m resolution grid in the horizontal dimension, and with 3 layers in the vertical direction (one upper soil layer, one lower soil layer and the groundwater layer). Some of these cells are hillslope cells, others are partially hillslope and partially river network cells, depending on the river width. The river width has been set in these cells between a minimum of 10 m and a maximum of 48 m, proportionally to the upstream area of the cell (the min and max widths are derived from cross section measurements provided by the Swiss Federal Office of the Environment). The river network cells are identified in the DEM by means of a flow accumulation routine in the preprocessing phase, and the initiation of the river network is set by the drainage area RT threshold (see point 2 above).

*- p5 l107: " catchment scale ": it is not precise enough. What scale?*

*- p5: put the dimensions of the variables presented in the equations. I do not understand the distinction between hillslopes and rivers in terms of erosion and transport processes. What is the link between the terms D and E?*

The erosion and transport processes on hillslopes and channels are clarified in point (2). D and E are not related to each other. D represents the erosion or deposition of sediment on the hillslopes, while E is the flux of sediments between the water column and the river bed in the river network.

*- p5 Eq.4: I do not understand the definition of X. It should be a width rather than a length for the calculation of the flux.*

X is the length of the river cell. Eq (4) is the integration over the longitudinal dimension of Eq (3), which is a 1D equation and therefore is already integrated over the cell width.

*- p7 l166-167: is this a wash load hypothesis?*

Yes, it can be described as such. However, we estimate that sediment transported in suspension in this catchment is between clay and medium sand grain size, therefore it includes also rather coarse grain sizes.

*- p8 Fig3 : SSC values seem low for a mountainous catchment area. This is certainly related to the lack of observed values during floods.*

Indeed, the bulk of observed SSCs are not very high (less than 20 mg/l) but during floods they can be much higher. We will discuss this it as proposed in point (3).

*- p8 l186-188: it is questionable to use the slope of the Q-SSC relation given the dispersion that exists between these two variables (even in log scale)*

We aim at reproducing the Q-SSC relation, as representative of the basin sediment dynamics, by matching the modelled and observed clouds of points, i.e. their trend and their dispersion, by looking at the SSC frequency distribution. In the revised manuscript, we will clarify that we did not match the slope of regression lines of observations and simulations, rather we looked at both the trend and the dispersion. In the revised manuscript, we will add the percentage of simulated SSC that fall within the observed percentiles, as proposed in the reply to Reviewer 1.

*- p10 l226-228: what forms of erosion are observed within the basin?*

Deep, permanent gullies and small shallow landslides characterize soil erosion in the eastern part of the basin (Fontanne catchment). The southeastern region is characterized by shallower gullies and scree, the major landslides are also located in this area and have a significant role in the sediment budget of the basin (Norton et al., 2008, Van der Berg et al., 2012).

*- p10 l234: "Hinderer et al. (2013)" is not present in the reference list.*

Hinderer et al. (2013) is in the reference list in p20 l 475-477.

*- p10 l237: "The underestimation of sediment load (...) we do not like to reproduce the largest measured sediment concentrations". This is a working hypothesis that should be placed in « Material and Method ».*

We will explain better that since we underestimate highest hourly SSCs we also underestimate annual sediment loads and therefore cannot compare our yields directly with other estimates in the literature.

*- p10 Fig5: indicate the observed data as red dots on the SSC time series.*

We will consider this option.

*- p11 Fig6(a): over which periods are the intensities calculated: over the rain periods only or over the whole period of simulation?*

Intensities are calculated over the entire simulation period.

*- p11 l242: I suggest modifying "where SIM 2 and 3 are compared respectively with SM1".*

We agree.

*- p13 Fig8(b): There is a black dot without a text caption*

This is T3, we will fix this.

*- p17 l369-372: I am not convinced by this hypothesis, which depends heavily on the nature of the soils and the infiltration model used.*

We will further discuss this point according to the results of point (7).

References

Mahoney, D.T., Fox, J.F., Al Aamery, N., 2018. Watershed erosion modeling using the probability of sediment connectivity in a gently rolling system. J. Hydrol. 561, 862–883, https://doi.org/10.1016/j.jhydrol.2018.04.034.

Heckmann, T., Cavalli, M., Cerdan, O., Foerster, S., Javaux, M., Lode, E., Smetanová, A.,Vericat, D., Brardinoni, F., 2018. Indices of sediment connectivity: opportunities, challenges and limitations. Earth Sci. Rev. 187, 77-108.

Norton, K. P., von Blanckenburg, F., Schlunegger, F., Schwab, M., and Kubik, P. W.: Cosmogenic nuclide-based investigation of spatial erosion and hillslope channel coupling in the transient foreland of the Swiss Alps, Geomorphology, 95, 474–486, https://doi.org/10.1016/j.geomorph.2007.07.013, 2008.

Van Den Berg, F., Schlunegger, F., Akçar, N., and Kubik, P.: 10Be-derived assessment of accelerated erosion in a glacially conditioned inner gorge, Entlebuch, Central Alps of Switzerland, Earth Surface Processes and Landforms, 37, 1176–1188, https://doi.org/10.1002/esp.3237, 2012.

---

## Author Response (AR1)

Dear Editor,

Please find enclosed the response to the reviewers' comments and a marked-up version of the manuscript. The manuscript has been revised according to the comments of the two reviewers and the short comment by Erkan Instanbulluoglu. In addition, we took the chance to improve the text and the order of some sections for better readability. We would like to thank the Editor,

5 the Associate Editor and the two anonymous reviewers for their constructive comments and suggestions, which helped us to improve the manuscript.

In the following, we report the text of the review in blue italic, and in black our reply.

**Response to Referee 1**

15

The manuscript describes a new suspended sediment flux model which is then used to analyze sediment dynamics and sources
in a mid-size catchment. The paper is well written. Unfortunately, I have great reservations about the model novelty and the interpretation of the results. I recommend Major Revisions as I think that the manuscript can be of interest once it is more properly framed.

General comments: The authors all but ignored existing large-scale sediment flux models (see a most relevant review paper and a couple of examples below). This is a major emission that must be corrected; their model should be framed in reference to these models.

In the submitted manuscript we indeed referenced only those models that in our opinion were directly comparable to the approach we took, like tRIBS (Francipane et al., 2012) and the model of Tsuruta et al. (2018). As the reviewer suggests, we added a reference to the large-scale sediment flux models WBMsed (Cohen et al., 2013) and Pelletier (2012) in line 69 of the revised manuscript and we highlighted in particular one of the main differences between these approaches and our model,

- 20 which is the lack of an explicit hydrology component. At the same time, we prefer to stay focused in the paper on physics-based modelling approaches only. Most of the models reviewed or presented in the papers suggested by the referee (De Vente et al., 2013; Cohen et al., 2013; Pelletier, 2012; Syvitski and Milliman, 2007) are statistical and steady state models which cannot be seen as reference models, because they are developed for a different purpose and cannot answer the same questions we address in our work.
- 25 The authors greatly over-sell the novelty and capabilities of the sediment model. While it is true that the hydrological framework is physically-based, the sediment model is a simple empirical equation (Eq. 2) that predicts sediment as a function of discharge, slope and a spatially variable (alpha) coefficient. Alpha is calibrated using USLE parameter combination. Sediment transport (Eq. 4) is a simple cell-to-cell and time-step balance. I see very little novelty in this model. The authors must make the argument of why this model is novel if they wish to continue claiming it.
- 30 This point and criticism has been addressed in lines 78-85 of the revised manuscript, where we highlight that the novelty consists in the combined hydrology-sediment system approach and the questions it allows to address. In particular, the novelty we perceive is based on the combination of the following elements: (a) We combine physically-based unsteady hydrological simulation of surface overland flow with a simple hillslope erosion and sediment transport component. This ensures that sediment is produced and transported along hillslopes by overland flow respecting physical processes of hillslope erosion and
- 35 sediment transport as we understand them. The sediment component is simple by design (sediment production and continuity in Eq. 2 and 4 mentioned above), so that the most uncertain part of the modelling system is not over-parameterized. (b) The high spatial and temporal resolutions of the model (100 m and 1 hr) allow the inclusion of detailed topographic variations, the explicit simulation of connectivity of sediment pathways in space and time, and the modelling of fast response to heavy precipitation where it happens. (c) Continuous simulation (order of decades) by our approach, allows to track overland runoff
- 40 generation and thus hillslope sediment transport, by spatially distributed simulation of the dynamics of soil moisture, snowmelt,

and rainfall, not only for individual events, but over long periods of time reflecting also long-term changes of key drivers of runoff generation mechanisms (e.g. soil moisture states, rainfall seasonality, etc).

None of the physics-based models reviewed in De Vente et al. (2013) or mentioned in the introduction of the manuscript combines these three characteristics at a spatial scale comparable to our case study. This is the context in which we perceive

45 the novelty of our work, and which allows us to explore the effects of the spatial variability in catchment erodibility and rainfall with higher confidence.

At the same time, we recognize that our model is not novel in the sense that it is the first and only such model. For example, it is similar to tRIBS (Francipane et al., 2012), DHSVM (Doten et al., 2006) and the model of Tsuruta et al. (2018). However, the first two are not applicable to large catchments and long and continuous simulations at high resolutions due to computational demand, while our setup is computationally very efficient and applicable to medium and large-scale basins, and the latter is a

50 demand, while our setup is computationally very efficient and applicable to medium and coarser resolution model with less physical hillslope surface runoff generation routines.

In conclusion, we do think that our approach has unique strengths that allow us to explore the hydrology-sedimentology connections leading to sediment generation pathways at high resolutions, which other approaches do not have. This is the novelty in the approach.

**55 The evaluation of the sediment model is odd - referring to the relatively low scatter in the SSC-Q plot (Fig 3) as an argument for strong model performance. A standard model performance analysis is offered for the model's hydrological predictions (Table 1). It seems that the observed sediment is used for model calibration so we actually left with little knowledge about how well the model is doing.**

- Indeed the referee is correct that it is much easier to calibrate the hydrological part of the model than the sedimentological one, mainly because we do not have the data to do so. There is only one suspended sediment measurement point at the outlet of the basin where bi-weekly measurements are available for a reasonably long period. We do not consider it meaningful to tweak the simple advection-based sediment transport routine implemented in the model, to match "perfectly" the observed hourly concentrations at the outlet measurements. Rather we assumed as a qualitative measure of success the reproduction
- of properties of the observed sediment rating curve (SRC), i.e. the relationship between hourly discharge and suspended sediment concentration (SSC), which captures the catchment sediment dynamics. Concretely, we calibrate the sediment model parameters, i.e. the river initiation threshold RT and the  $\alpha_1$  erodibility parameter (Eq. 5), to reproduce (a) the observed SSC-Q cloud of point as best as we can, as well as (b) the frequency distribution of observed SSCs. We also omitted a traditional validation with a part of the dataset not used in calibration as our observed records are too short and our main focus is on the
- 70 sensitivity to input data (spatial variability in surface erodibility and rainfall) not on the predictive uncertainty in SSC per se. In the revised paper we rephrased section 2.3.3 (section 2.2.3 of the submitted manuscript) to clarify the method used (in particular, see lines 216-218) and we introduced the percentage of modelled SSCs that fall within the  $5^{th}$  and  $95^{th}$  percentile of the observations in SIM 1, as a quantitative indicator of the model performance (see lines 224-225).

**Given the relative simplicity of the model and the way it was calibrated, the interpretation of the model results extends much beyond the model's ability to represent the discussed processes. The authors need to frame their analysis within the model's capabilities to represent the relevant processes and drivers. Some examples of overreaching are 1st sentence in the Discussion, sentence starting in lines 300, 311 and 315.**

We respectfully disagree with the referee about this point. The spread around the SRC in our deterministic approach is due to (a) the spatially distributed nature of the model, which allows to simulate the heterogeneous response of the basin to hydrological forcing, based on the topographic characteristics, depth and properties of the soil, (b) the spatial variability of surface erodibility and the connectivity of hillslope flow paths to the river network, and (c) the spatio-temporal distribution

of rainfall leading to overland flow and erosion (lines 293-302 of the submitted manuscript). We are of the opinion that all of these processes are robustly included in our modelling approach. Of course they cannot explain all the SRC spread because in the real natural catchments there is an added element of stochasticity in sediment mobilization, transport, and sediment supply

85 limitations, which add to the SRC variability (lines 306-312 of the submitted manuscript). However, we believe that this does

not invalidate our modelling results or their ability to provide insights into process controls, like the role of spatial variability in erosion drivers.

In the revised manuscript we have clarified the limitations of the results in lines 340-343 and 447-451. We also rephrased the research questions, as suggested by Reviewer 2, to be more specific on the conclusions we can draw from the four simulated scenarios.

90 sce

**Response to Referee 2**

115 The paper presents a hydrosedimentary model that couples the TOPKAPI-ETH hydrological model to a physically based and spatially distributed erosion and sediment transport model. [...]

However, the questions asked are not precise enough. It is a bit ambitious to want to answer such generic questions with only 4 scenarios. The model is very little evaluated in terms of erosion before analysing the results of the different scenarios. It is therefore difficult to give credit to the results obtained. My advice would be to reformulate questions that are compatible

with the framework offered by the tested scenarios and to rework the results and discussion sections according to these new 120 questions.

**I therefore recommend major revisions for this paper.**

We modified the research questions in order to be more specific on the analyses that we performed, i.e. the investigation about the location of sediment sources, their productivity and connectivity to the river network and how these information help to explain the sediment load observed at the outlet (lines 90-98). Moreover, we further supported some of the discussion statements with additional analyses of the hydrological results, as suggested by the reviewer.

General remarks:

125

- The authors do not mention the DHSVM model although it would be a very relevant tool for this type of catchment. It is necessary to justify the development of a new model compared to existing models such as DHSVM.
- (1) DHSVM is indeed a relevant tool in the framework of these type of models. DHSVM features a rigorous description 130 of the hydrological processes and the role of vegetation and simulates sediment production by hillslope erosion, road erosion and mass wasting. TOPKAPI-ETH presents a slightly more simplified description of the hydrological processes and only includes erosion by overland flow. These choices are aimed at avoiding over-parameterization of the model and at keeping it computationally efficient and thus suitable for mesoscale and large catchment applications and small grid sizes, even when
- 135 further components would be added (e.g. additional sediment transport processes). We introduced the DHSVM model in the literature review (lines 71-72) and highlighted the differences and novelties of our model in lines 72-73 and 76-85.

**- The description of the erosion model did not seem clear enough to me, especially the distinction between the representation of hillslopes and river processes.**

140 (2) Sediment production and transport on the hillslopes is based on a transport capacity-mass balance approach, i.e. sediment flux is assumed to be always at transport capacity, and the model simulates erosion or deposition when there is a change in transport capacity. Sediment delivered to the channel network is advected by the river flow, there is no possibility for deposition or entrainment of fine sediment from the bed in the channel. The transition between sediment transport description as hillslope process to channel process corresponds to the transition from water flow routing as overland flow to channelized flow. This 145 takes place between hillslope and channel cells and is fundamentally determined by the drainage area threshold (RT) used to identify river cells in the DEM, i.e. the river initiation threshold, which was a parameter in our model.

We clarified in section 2.1 the distinction between the description of sediment processes on the hillslopes and in the channel (lines 119-120 and 128-131).

**150 - The authors use data from Swiss operational services. However the temporal frequency of SSC data is too low for a catchment of this size located in a mountainous area. Flood events are most likely under-sampled. High SSC values are probably missing from the data set for this reason.**

(3) We agree with the reviewer on this point and we explicitly stated this limitation of the data in in lines 159-161 of the revised manuscript. There is nothing we can do about the temporal frequency of the data (twice a week). All sediment monitoring stations of the Federal Office of the Environment in Switzerland have such low frequency except for few automatic

155

stations with turbidity measurements in recent years. In lines 214-216 of the revised manuscript we clarified that in the calibration of the model we focused only on the lowest  $85^{th}$  percentile of the SSC dataset. This choice is motivated both by the under-sampling of extreme SSCs in the data and by the expected underestimation of high SSCs by the model, given that very localized sediment sources are not simulated.

160

**- One could be interested in the impact of the scenarios on the hydrological response and the indirect impact this may have on sediment dynamics.**

(4) This is a good point. We added Figure 6, where the hydrological response in the 4 simulations is compared, by means of the mean annual flow Qmean, the annual flood Qmax and the variability of flow at the basin outlet, expressed by the coefficient of variation *CV*, and the mean annual surface runoff on the hillslopes QOFmean. The plots in Figure 6 explain the higher sediment production in SIM 1 and 3, by the greater runoff production indicated by Qmean, Qmax and QOFmean (line 305-307), and shows that spatial variability of precipitation is a source of flow variability, and thus favors SSC-Q scatter (lines 347-348).

**170 - *The concepts of structural and functional connectivity, widely present in the literature, are not discussed although they are at the heart of the subject developed in the paper.**

- Connectivity indices are not used.

Both comments have been addressed in point (6) below.

**175 - The process of detachment by rain is not taken into account in the model. Only the process of detachment by runoff is taken into account. This is questionable when the objective is to estimate the effect of spatial variability of precipitation.**

(5) Detachment by rain, together with the overland flow entrainment capacity, defines the amount of sediment available for transport. In our model, we do not simulate the local rainsplash detachment and overland flow entrainment separately from the mobilization processes at the grid scale, rather we assume that sediment on the hillslopes is always available to fulfill the overland transport capacity. Sediment availability is only limited by the soil depth (sediment layer thickness), however, this limitation does not play a role in our simulations. It would be necessary to include the process of rainfall detachment by rainsplash if the model distinguished between the processes of sediment detachment at the microscale, determining the sediment available for transport, and sediment mobilization.

We agree that the submitted manuscript is unclear in this regard. In the revised manuscript we replaced "sediment production" with "sediment mobilization" at several points and we rephrased lines 119-120, to clarify that we model the maximum amount of sediment that overland flow can transport, and not the amount detached by rainfall processes or overland flow entrainment.

**- Connectivity index maps could be used to study the spatial organization of erosion (Section 4.2). It is questionable whether there is any real added value in using the model presented in this study to address this issue.**

- (6) The vast majority of connectivity indices provides a static description of the structural or functional sediment connectivity, based on the upslope contributing area as a proxy for discharge to estimate stream power (Heckmann et al., 2018). The sediment delivery ratio SDR simulated by our model quantifies the proportion of eroded sediments that are routed to a point on a river network or outlet of a selected subbasin, by action of overland flow and channel flow. As such, SDR is a dynamic indicator of functional connectivity, where the discharge (and thus stream power) is considered explicitly as a function of the temporal and spatial variability of the hydrological forcing and topographic characteristics, instead of being represented by the
- upstream area only. In fact, besides accounting for the time dependency of discharge, SDR also integrates the variability in space of the functional connectivity, by substituting the unique Q-A relationship used in traditional connectivity indices, with the explicit simulation of overland flow on the hillslopes.

180

200

205

We have applied this analysis at the subbasin scale, but SDR could be potentially computed for each grid cell to build a connectivity map. Mahoney (2017) propose a comparable approach that quantifies dynamic functional connectivity, based on hydrological modelling too. We added this discussion points to complement section 5.2 (lines 380-386).

**- In section 3.4, the authors examine the results at the temporal scale of the flood event. It is difficult to examine the effect of soil moisture on erosion and sediment transport without giving guarantees on the performance of the model in reproducing flows under dry and wet conditions.**

(7) We added a comparison between the hydrological model performance for the low initial soil moisture (SM0) events and the high SM0 ones, which are analysed in section 3.4 of the submitted manuscript. This comparison is presented in Table S2 of the revised manuscript with the performance indices, and in Figure S3 as a density plot, and it shows that the model tends to overestimate the flow in both types of events, but especially at low initial soil moistures. Based on the findings of Paschalis et al. (2014) and Shah et al. (1996), we do expect to see an effect of initial soil moisture on erosion and sediment transport, as it is suggested by Figure 10b of the revised manuscript. However, we also note that our results do not allow for a clear conclusion, given the small difference between the sediment load distributions of low and high SM0 events and the tendency to overestimate flow in low SM0 events. We modified the discussion of Figure 10b according to these observations (lines 433-438).

215

**- The summary at the beginning of Section 4.3 is interesting.**

We thank the reviewer for the appreciation. It is indeed an important part to link together the catchment-wide analysis of the sediment dynamics with the sediment signal at the outlet.

Specific remarks :

220 - p2 133: I would suggest adding "especially in small to medium catchments (up to 1000 km2) after "the strong nonuniqueness of suspended sediment concentrations (SSCs)".

We agree with the suggestion and we added it in line 38-39.

- p2 137: I would suggest adding "and transfers " after "in sediment mobilization".

We agree with the suggestion and we added it in line 43.

We rewrote this part, now introducing the concepts of structural and functional connectivity, and the indices of sediment connectivity (lines 44-58).

- p2 l48: add reference Misset et al (2018)

Misset C., Recking A., Legout C., Poirel A., Cazilhac M., Esteves Michel, Bertrand M. (2019). An attempt to link suspended load hysteresis patterns and sediment sources configuration in alpine catchments. Journal of Hydrology, 576, 72-84. ISSN 0022-1694

235

Thanks for suggesting this reference, it is now included in line 48.

- p2 l56: replace "transport" by "transfer" in several places in the text

- *p2 l39 to 52: rewrite this part which is not clear and take into account the concepts of structural and functional connec-tivity.*

We agree with this comments and we now replaced "transport" with "transfer", where "transport" indicated the group of processes transferring sediments from the sources to the outlet. We kept the word "transport" to indicate the specific processes of sediment transport in the hillslope or channel cells.

- p3 l65: replace "cesar-lisflood" with "caesar-lisflood".

Thanks, we corrected this typo.

245

- p3 l62 to 75: add the reference to the DHSVM model and explain the added value of the model presented in relation to this model

See point (1).

250 - p3 l77: "a physically explicit spatially distributed deterministic model": simplify the formula. What does "explicit" mean here?

"Explicit" means that the model is based on a physical representation of most processes, but it still contains some conceptualisations or approximations of the processes. For general understanding, we replaced "explicit" with "based" in the revised manuscript.

255

- p3 183: "mean annual discharge" instead of "average discharge".

We agree and modified it in line 146.

- p3 l87 : " mostly driven by overland flow ". What about rainfall processes ?

260 This is correct, we modified the sentence in line 150.

- p4 l97: what is the scale for the soil map?

The scale is given by the coordinates on the x- and y- axes (units are the same as in Figure 1a).

**265 - *p4* 1103: Is it really 2D whereas the equations presented p5 are 1D?**

The solution is 1D in the direction of the steepest descent at the grid scale for surface and subsurface flow. All inflows from the neighbourhood cells are integrated in space. For clarity, we removed the "2D" from line 110.

**p5* 1105 to 114: I do not understand how the hydrographic network is represented and discretized. The same question applies to the hillslopes. A specific part is missing for describing the discretization used in the model.**

The entire basin is discretized as a 100 m resolution grid in the horizontal dimension, and with 3 layers in the vertical direction (one upper soil layer, one lower soil layer and the groundwater layer). Some of these cells are hillslope cells, others are partially hillslope and partially river network cells, depending on the river width. The river width has been set in these cells between a minimum of 10 m and a maximum of 48 m, proportionally to the upstream area of the cell as in the downstream

275 hydraulic geometry relations of Leopold and Maddock (the min and max widths are derived from cross section measurements provided by the Swiss Federal Office of the Environment). The river network cells are identified in the DEM by means of a flow accumulation routine in the preprocessing phase, and the initiation of the river network is set by the drainage area RT threshold (see point (2) above). In the revised manuscript, we added this description of the model discretization between lines 104 and 109.

280

**- p5 l107: " catchment scale ": it is not precise enough. What scale?**

The model allows high resolution simulations in catchments up to large scales (>1000 km2). We added this information in line 116.

**285 - *p5*: put the dimensions of the variables presented in the equations. I do not understand the distinction between hillslopes and rivers in terms of erosion and transport processes. What is the link between the terms D and E?**

The erosion and transport processes on hillslopes and channels are clarified in point (2). D and E are not related to each other. D represents the erosion or deposition of sediment on the hillslopes, while E is the flux of sediments between the water column and the river bed in the river network. We added the dimensions of the variables in section 2.1.

290

**- p5 Eq.4: I do not understand the definition of X. It should be a width rather than a length for the calculation of the flux.**

X is the length of the river cell. Eq. (4) is the integration over the "along-flow" dimension of Eq. (3), which is a 1D equation and therefore is already integrated over the cell width.

**295 - *p7 l*166-167: *is this a wash load hypothesis?**

Yes, it can be described as such. However, we estimate that sediment transported in suspension in this catchment is between clay and medium sand grain size, therefore it includes also rather coarse grain sizes.

**- p8 Fig3 : SSC values seem low for a mountainous catchment area. This is certainly related to the lack of observed values during floods.**

Indeed, the bulk of observed SSCs are not very high (less than 20 mg/l) but during floods they can be much higher. In the revised manuscript we discussed this limitation in lines 159-161 and 214-216 (see point (3)).

**- p8 l186-188: it is questionable to use the slope of the Q-SSC relation given the dispersion that exists between these two variables (even in log scale)**

We aim at reproducing the Q-SSC relation, as representative of the basin overall sediment dynamics, by matching the modelled and observed clouds of points, i.e. their relation and dispersion, by looking at the SSC frequency distribution. In the revised manuscript, we rephrased the description of the calibration procedure (see lines 216-218), to clarify that we did not match the slope of regression lines of observations and simulations, rather we looked at both the trend in the relation and the dispersion. In line 224-225, we also added the percentage of simulated SSC that fall within the observed percentiles, as discussed in the reply to Reviewer 1.

310

300

**- p10 l226-228: what forms of erosion are observed within the basin?**

Deep, permanent gullies and small shallow landslides characterize soil erosion in the northwestern part of the basin (Fontanne 315 catchment). The southeastern region is characterized by shallower gullies and scree, the major landslides are also located in this area and have a significant role in the sediment budget of the basin (Norton et al., 2008; Van Den Berg et al., 2012). In lines 271-272 of the revised manuscript we added a reference to the discussion section 5.2, where the geomorphological differences between the two regions of the basin are described.

*p10 l234: "Hinderer et al. (2013)" is not present in the reference list.*

Hinderer et al. (2013) is in the reference list of the submitted manuscript at lines 475-477.

- p10 l237: "The underestimation of sediment load (...) we do not like to reproduce the largest measured sediment concentrations". This is a working hypothesis that should be placed in « Material and Method ».
- We modified the sentence to explain better that, since we underestimate highest hourly SSCs, we also underestimate annual sediment loads and therefore underestimate the yield estimates found in the literature (lines 279-281 of the revised manuscript).
  - p10 Fig5: indicate the observed data as red dots on the SSC time series.

Thanks for the suggestion, the observed data have been added to the SSC time series in Figure 5a.

330

- *p11 Fig6(a): over which periods are the intensities calculated: over the rain periods only or over the whole period of simulation?*

Intensities are calculated over the entire simulation period.

**335 - p11 l242: I suggest modifying "where SIM 2 and 3 are compared respectively with SM1".**

We agree and we modified the sentence (line 292).

- *p13 Fig8(b): There is a black dot without a text caption*

This is T3, we fixed it.

**340**

- *p17 l369-372: I am not convinced by this hypothesis, which depends heavily on the nature of the soils and the infiltration model used.*

We modified this discussion point (lines 408-413) based on the hydrological performance of the model for the low and high initial soil moisture events and highlight that our results hint to a greater sensitivity of sediment load to precipitation spatial distribution when the initial  $SM_0$  is low, but that at the same time they do not allow for a clear conclusion (see point (7) above).

[revised manuscript text omitted]

---

## Author Response (AR2)

Dear Editor,

Please find enclosed the response to the associate editor's and reviewer's comments and a marked-up version of the manuscript. To address the two main issues raised by the reviewer and the editor, the phrasing of some sentences has been modified in the main text and we have added an additional model performance evaluation in the supplementary material. We would like to thank the Editor, the Associate Editor and the anonymous reviewer for their comments.

In the following, we report the text of the review in blue italic, and in black our reply.

**Response to Editor**

*The authors have done good job at addressing the review comments. This has been recognised by the Reviewer. However, I agree with the Reviewer that there are several issues that raise some concerns. (1) the novelty of the model and approach and (2) the validation of the model. These issues are required to be addressed before the paper can be published. Addressingb these issues will make the paper stronger and likely have higher impact.*

The novelty of the model is presented in lines 76 to 85 of the manuscript and consists in the combination of three key features, which are essential for a physically-based representation of the processes of soil erosion and suspended sediment transport, and are only partially included in the models available in the literature so far. Such features are (1) the combination of unsteady spatially distributed simulation of surface and subsurface water fluxes with a hillslope erosion and sediment transport component, which is simple to avoid over-parameterization and to maintain computational efficiency, which enables applications to mesoscale catchments, (2) the possibility to run continuous high spatial ($\Delta x$=100 m) and (3) temporal resolution ($\Delta t$=1 hr) simulations, to represent the hydrological drivers of erosion based on the local changes in soil moisture dynamics, to capture fast runoff response to precipitation and the connectivity of water and sediment pathways in the catchment over time. We have added an additional statement in lines 85-87 to highlight the relevance of the combination of such features as the novelty of our model.

To address the concern about the validation of the model, we have followed the suggestion of the reviewer to use sediment rating curves to extrapolate a continuous time series of suspended sediment concentration (SSC) from intermittent observations. In the reply to the reviewer we present the results of the comparison of the extrapolated observed time series with the simulated one. We agree that this evaluation can strengthen the presentation of the sediment module, therefore we have added it to the supplementary material.

**Response to Referee**

*The revised manuscript address most of the issues and critics raised by myself and the other reviewer. The two main issues I had with the original manuscript were the lack of novelty and improper validation. The authors addressed the first issue by*
30 *explicitly describing the model novelty. They make a good argument and now present the model in a more realistic way. I still do not find the model to be very novel, but I am satisfied with the new framing of it.*

We have further strengthened the presentation of the model, by underlining that its novelty consists in the combination of three very important features for the modelling of soil erosion and sediment transport, which allow us to address the research questions we raised in the paper (line 85-87) - see also reply to the editor. In our opinion the combined hydrology-sediment
35 model is unique in its process completeness and applicability to mesoscale catchment simulation at high resolutions.

*As for the model validation, the authors justify the use of Q-SSC plots by the limitation in observed sediment stating: "There is only one suspended sediment measurement point at the outlet of the basin where bi-weekly measurements are available for a reasonably long period." Limited sediment observations are nearly always the case. Bi-weekly measurements for an extended period are actually quite a lot for this size basin. Indeed, the authors added observed SSC data to Figure 5 but offer only*
40 *qualitative interpretation. There are several approaches for validating the model with this data. Comparing the model for the days the data was collected, yearly or seasonal averages, and rating curves with discharge can be derived to generate continues timeseries. The latter is basically how it is done in sediment stream gaging.*

We agree with the reviewer that the performance of the suspended sediment module can be further evaluated by one of the mentioned approaches. We followed the suggestion and we present here the results. By extrapolating a sediment rating curve
45 from the observations, we obtained a continuous time series of hourly SSC from the observations by feeding the rating curve with the observed hourly discharge. We have assessed the performance of the model in reproducing such extrapolated observed SSC, by limiting our analysis to the SSCs lower than the $85^{th}$ percentile of the observed SSC distribution, coherently with the approach applied in the calibration procedure (see section 2.3.3). We considered four time aggregations (hourly, daily, monthly and yearly) and computed the following indices: correlation coefficient r, percent bias PBIAS, normalized root mean square
50 error nRMSE and mean absolute error MAE. As expected, the results (Table 1 and Figure 1) show a tendency to underestimate the observations and a lower performance at the hourly scale where the sediment routing procedure becomes critical. However, the improved performance with increasing temporal aggregation demonstrates that the hydrological drivers of erosion and sediment transport in the model are properly represented, especially at temporal scales typical of the hydrological processes.
Table 1 and Figure 1 have been added in the supplementary material of the manuscript.

**Table 1.** Performance of the suspended sediment simulation for the period 2004-2016 at the outlet of the river basin, in terms of correlation coefficient (r), Percent Bias (PBIAS), normalized root mean square error (RMSE) and mean absolute error (MAE) for data simulated at the hourly resolution and aggregated to daily, monthly and annual values. The analyses has been limited to the values lower than the $85^{th}$ percentile of the observations.

|  | r | PBIAS | nRMSE | MAE |
|---|---|---|---|---|
|  | [-] | [%] | [-] | [mg/l] |
| Hour | 0.51 | -12.14 | 1.02 | 8.86 |
| Day | 0.52 | -12.14 | 0.91 | 9.18 |
| Month | 0.64 | -12.14 | 0.52 | 5.64 |
| Year | 0.52 | -12.14 | 0.19 | 2.28 |

55 *The model description is was much improved in the revised manuscript. The in-channel equations (Eq. 3 and 4) could be further clarified with a simple example which will give the reader a sense of the value space and how the variables are derived.*

Eq. 3 is a simple advection equation, and Eq. 4 has been derived by integrating Eq. 3 along the length of the grid cell $X$, as explained in line 138. For example, the term $\frac{\partial AC}{\partial t}$ corresponds to $\frac{\partial V_i C_i}{\partial t}$, and $E$ to $E_i X$. For very similar examples of such

[Figure]

**Figure 1.** Density plot of simulated vs observed hourly suspended sediment concentrations at the outlet of the river basin for the period 2004-2016. The black dashed line indicates the $85^{th}$ percentile of the observations, to which the performance assessment has been limited.

60 integration applied to the equations of the hydrological component of the model, the reader is referred to Liu and Todini (2002) (see line 137 of the paper).

Eq. 3 and 4 contain two parameters: the grid cell size $X$ (Eq. 4), which corresponds to the spatial resolution of the model ($\Delta x$=100 m, see line 176), and the bed-exchange term $E$, which we set to 0 as explained in line 203. The variables contained in the equations are defined between line 136 and 143, and in the model are the result of the hydrological and transport processes 65 simulated in each cell. Therefore we think that it would not be very significant to report the values for one specific cell at one time step of the simulation.

*The authors state that "...this model offers a valuable tool for investigating..."; is the model freely available?*

The executable of the model is currently available in an experimental version and can be requested to the authors. A new version is currently being finalized and will be presented later this year.

[revised manuscript text omitted]